# Dual engagement of the nucleosomal acidic patches is essential for deposition of histone H2A.Z by SWR1C

Alexander S Baier[1,2], Nathan Gioacchini[1,3], Priit Eek[4,5], Erik M Leith[4], Song Tan[4], Craig L Peterson[1]*

[1]Program in Molecular Medicine, University of Massachusetts Chan Medical School, Worcester, United States; [2]Medical Scientist Training Program, T.H. Chan School of Medicine, University of Massachusetts, Boston, United States; [3]Interdisciplinary Graduate Program, Morningside Graduate School of Biomedical Sciences, University of Massachusetts Chan Medical School, Worcester, United States; [4]Center for Eukaryotic Gene Regulation, Department of Biochemistry and Molecular Biology, Pennsylvania State University, University Park, United States; [5]Department of Chemistry and Biotechnology, Tallinn University of Technology, Tallinn, Estonia

*For correspondence:
Craig.Peterson@umassmed.edu

Competing interest: The authors declare that no competing interests exist.

**Abstract** The yeast SWR1C chromatin remodeling enzyme catalyzes the ATP-dependent exchange of nucleosomal histone H2A for the histone variant H2A.Z, a key variant involved in a multitude of nuclear functions. How the 14-subunit SWR1C engages the nucleosomal substrate remains largely unknown. Studies on the ISWI, CHD1, and SWI/SNF families of chromatin remodeling enzymes have demonstrated key roles for the nucleosomal acidic patch for remodeling activity, however a role for this nucleosomal epitope in nucleosome editing by SWR1C has not been tested. Here, we employ a variety of biochemical assays to demonstrate an essential role for the acidic patch in the H2A.Z exchange reaction. Utilizing asymmetrically assembled nucleosomes, we demonstrate that the acidic patches on each face of the nucleosome are required for SWR1C-mediated dimer exchange, suggesting SWR1C engages the nucleosome in a 'pincer-like' conformation, engaging both patches simultaneously. Loss of a single acidic patch results in loss of high affinity nucleosome binding and nucleosomal stimulation of ATPase activity. We identify a conserved arginine-rich motif within the Swc5 subunit that binds the acidic patch and is key for dimer exchange activity. In addition, our cryoEM structure of a Swc5–nucleosome complex suggests that promoter proximal, histone H2B ubiquitylation may regulate H2A.Z deposition. Together these findings provide new insights into how SWR1C engages its nucleosomal substrate to promote efficient H2A.Z deposition.

## eLife assessment

This manuscript presents an **important** analysis of the role that the nucleosome acidic patch plays in SWR1-catalyzed histone exchange. This manuscript contains **convincing** data which significantly expands our understanding of the complex process of H2A.Z deposition by SWR1 and therefore would be of interest to a broad readership.

## Introduction

Eukaryotic genomes regulate access to information stored in their genetic code through the dynamic nucleoprotein structure known as chromatin. The basic unit of chromatin, the nucleosome,

sequesters ~147 base pairs (bp) of DNA around a histone octamer consisting of a H3–H4 tetramer flanked by two H2A–H2B (AB) heterodimers (*Luger et al., 1997*). Rather than a static structure, chromatin is highly dynamic, due to the coordinated impact of histone post-translational modifications, ATP-dependent chromatin remodeling enzymes, and the incorporation of histone variants (*Clapier et al., 2017*). Disruption of these processes can have deleterious consequences for all nuclear events, including transcription, DNA replication, and genome stability pathways.

H2A.Z, a variant of the core histone H2A, is enriched within nucleosomes adjacent to genes transcribed by RNA polymerase II, replication origins, centromeres, and at sites of DNA double strand break repair (*Albert et al., 2007*; *Barski et al., 2007*; *Hartley and Madhani, 2009*; *Boyarchuk et al., 2011*; *Xu et al., 2012*). In yeast, H2A.Z enrichment correlates with higher rates of nucleosome turnover that facilitates transcription, as well as antagonizing the spread of heterochromatic regions from telomeres and the silent mating type loci (*Meneghini et al., 2003*; *Guillemette et al., 2005*; *Dion et al., 2007*). Recently, we found that H2A.Z plays a global, positive transcriptional role in yeast strains lacking the nuclear RNA exosome (*Bryll and Peterson, 2022*). While yeast H2A.Z is not essential for viability, H2A.Z is essential in metazoans where it is associated with both transcriptional activation and repression (*Raisner et al., 2005*). For instance, in embryonic stem cells, deposition and function of H2A.Z are interdependent with the polycomb repressive complex 2 (PRC2), and loss of either H2A.Z or PRC2 leads to transcriptional de-repression of a host genes and a subsequent failure in lineage commitment (*Creyghton et al., 2008*).

The site-specific deposition of H2A.Z is catalyzed by ATP-dependent chromatin remodeling enzymes related to the 14-subunit, yeast SWR1C complex (*Kobor et al., 2004*; *Mizuguchi et al., 2004*). Mammals contain two enzymes related to SWR1C, the SRCAP and Tip60/p400 complexes, and subunits of the mammalian enzymes are highly conserved with those of SWR1C (*Johnston et al., 1999*; *Fuchs et al., 2001*). Each of these enzymes are members of the larger INO80 subfamily of chromatin remodeling enzymes, distinguished by a large insertion domain between the two ATPase lobes of the catalytic ATPase (Swr1 in SWR1C) (*Clapier et al., 2017*; *Ruhl et al., 2006*; *Luk et al., 2010*). This insertion domain serves as a docking site for several key subunits of SWR1C-like complexes, including the Rvb1/Rvb2 heterohexameric ring (RUVB1/RUVB2 in humans) which acts as a further scaffold for organizing additional subunits (*Wu et al., 2005*; *Willhoft et al., 2018*). In addition to Rvb1/Rvb2, previous studies have demonstrated that the deposition of H2A.Z by SWR1C requires several key subunits, including Swc2, Swc4, Yaf9, and Swc5 (*Wu et al., 2005*; *Watanabe et al., 2015*). The Swc2 subunit and its mammalian ortholog, Yl-1, promote nucleosome binding and assist SWR1C in the recognition of nucleosome-free regions adjacent to gene promoter regions (*Wu et al., 2005*; *Ranjan et al., 2013*; *Liang et al., 2016*). Swc4 (DMAP1 in mammals) is also part of the yeast NuA4 histone acetyltransferase complex, and it appears to promote DNA and histone binding (*Lu et al., 2009*; *Gómez-Zambrano et al., 2018*). Swc5 (Cfdp1 in mammals) is essential for the ATPase activity of SWR1C, but how Swc5 promotes SWR1C activity is poorly understood (*Sun and Luk, 2017*).

The subunit complexity of chromatin remodelers varies from the single subunit, yeast CHD1, to multi-subunit ~1 MDa enzymes such as RSC, SWI/SNF, INO80C, and SWR1C. Despite differences in function and complexity, recent studies have shown that the activity of many remodelers require that they engage a solvent exposed, acidic surface on the nucleosome (*Dann et al., 2017*; *Gamarra et al., 2018*; *Dao et al., 2020*; *Levendosky et al., 2016*; *Eustermann et al., 2018*; *Lehmann et al., 2020*). This 'acidic patch' is composed of eight residues from histones H2B and H2A that are known to provide a docking interface for many proteins. For instance, PRC1, RCC1, and Orc1 bind to the nucleosome acidic patch through a common binding motif consisting of a loop region with an arginine residue that inserts directly into the acidic pocket (*Makde et al., 2010*; *McGinty et al., 2014*; *De Ioannes et al., 2019*). Though these regions are primarily unstructured and basic, nucleosomal docking requires the specific side chain structure of an arginine, which cannot be substituted by lysine, resulting in these regions being referred to as 'arginine anchors' (*McGinty and Tan, 2015*; *McGinty and Tan, 2021*). In the context of remodelers, ISWI uses an arginine anchor within its catalytic ATPase subunit to relieve an autoinhibitory mechanism following nucleosome binding (*Gamarra et al., 2018*). Members of the SWI/SNF family of enzymes also use arginine-rich motifs within both the conserved Snf5- and Snf2-like subunits to sandwich the complex to the octamer during remodeling activity (*Han et al., 2020*; *Wagner et al., 2020*; *Ye et al., 2019*). A recent cryoEM structure of INO80C also shows key interactions with the nucleosomal acidic patch (*Eustermann et al., 2018*),

and this remodeler is unable to remodel nucleosomes that lack this nucleosomal epitope (*Gamarra et al., 2018*).

Here, we investigate the role of the nucleosome acidic patch in dimer exchange by SWR1C. Through Förster-resonance energy transfer (FRET), fluorescence intensity (FI), and fluorescence polarization (FP) assays, we interrogate how SWR1C is influenced by H2A/H2B acidic patch alterations, and we specifically probe how each acidic patch on the two faces of the nucleosome contribute to SWR1C activity. Surprisingly, we find that loss of even a single acidic patch results in the loss of dimer exchange activity, suggesting SWR1C must engage both acidic patches simultaneously in a 'pincer-like' fashion to carry out dimer exchange. In addition, we found that both linker DNA and the contra-lateral incorporation of an H2A.Z/H2B dimer-stimulated H2A.Z deposition. Efforts to identify potential SWR1C subunits responsible for acidic patch interactions led us to identify an arginine-rich motif within the Swc5 subunit of SWR1C, and we show that this domain is essential for dimer exchange in vitro, the function of SWR1C in vivo, and Swc5 binding to the nucleosome. A low-resolution cryoEM structure of the Swc5–nucleosome complex confirms an acidic patch interaction, as well as contacts with the histone H4 N-terminal tail and the H2B C-terminal helix. These results indicate that SWR1C activity relies on communication between the Swc5 arginine motif and the nucleosome acidic patch to functionally engage nucleosomes for dimer exchange activity.

## Results

### Dimer exchange by SWR1C requires the nucleosomal acidic patch

To investigate the role of the nucleosomal acidic patch on dimer exchange by SWR1C, a fluorescence-based assay was initially employed. Recombinant nucleosomes were assembled with octamers containing wildtype *Saccharomyces cerevisiae* H2A/H2B (AB) dimers and *Xenopus laevis* H3/H4 tetramers or with *S. cerevisiae* H2A/H2B dimers containing eight alanine substitutions for residues that comprise the nucleosomal acidic patch (AB-apm, see Method section for residue list). The DNA template for nucleosome reconstitutions was a 224-bp fragment containing a '601' nucleosome positioning sequence and an asymmetric, 77 bp linker DNA to mimic the structure of a promoter-proximal nucleosome located next to a nucleosome-depleted region (77N0) (*Lowary and Widom, 1998*). These substrates contain a Cy3 fluorophore conjugated to the linker distal end of the nucleosomal DNA, and a Cy5 fluorophore attached to an engineered cysteine residue (H2A-119) within the histone H2A C-terminal domain. The Cy3 and Cy5 fluorophores are within an appropriate distance to function as an FRET pair, such that excitation of the Cy3 donor leads to efficient energy transfer to the Cy5 acceptor, as evidenced by the fluorescence emission peak at 670 nm. The dimer exchange activity of SWR1C is monitored by following the decrease in the 670 nm FRET signal due to eviction of the Cy5-labeled AB-Cy5 dimer (*Figure 1A*).

Reactions to measure dimer exchange by SWR1C were performed under single turnover conditions (excess enzyme to nucleosomal substrate) and contained free H2A.Z/H2B (ZB) dimers which act as an essential co-substrate. Addition of SWR1C to a wildtype nucleosome led to a rapid drop in FRET, showing the biphasic kinetics consistent with the sequential exchange of the two nucleosomal H2A/H2B (AB) dimers (*Fan et al., 2022*; *Singh et al., 2019*). Incubation of SWR1C with a nucleosome that lacks an intact acidic patch (AB-apm/AB-apm nucleosome) showed little decrease in the FRET signal, indicating that an intact nucleosomal acidic patch is essential for SWR1C to catalyze H2A/H2B eviction (*Figure 1B*).

Since the FRET assay is limited to monitoring the eviction of nucleosomal H2A/H2B dimers, a gel-based assay that monitors deposition of H2A.Z was employed to confirm these findings (*Luk et al., 2010*). In this assay, an unlabeled, 77N0 nucleosome was incubated with SWR1C, ATP, and free H2A.Z/H2B dimers in which H2A.Z contains a 3xFLAG tag at its C-terminus. Reaction products are separated on native-PAGE, and formation of the heterotypic (AB/ZB) and homotypic (ZB/ZB) nucleosomal products are detected by their reduced gel migration due to the 3xFLAG tag on H2A.Z. Similar to the FRET-based assay, the SWR1C-catalyzed deposition of H2A.Z was largely complete by 30–60 min for the AB/AB nucleosomal substrate, but addition of SWR1C to the AB-apm/AB-apm nucleosome led to 10-fold reduction in H2A.Z deposition after 60 min (*Figure 1—figure supplement 1A, B, D*). Together, these assays indicate that SWR1C requires an intact nucleosomal acidic patch for H2A.Z deposition.

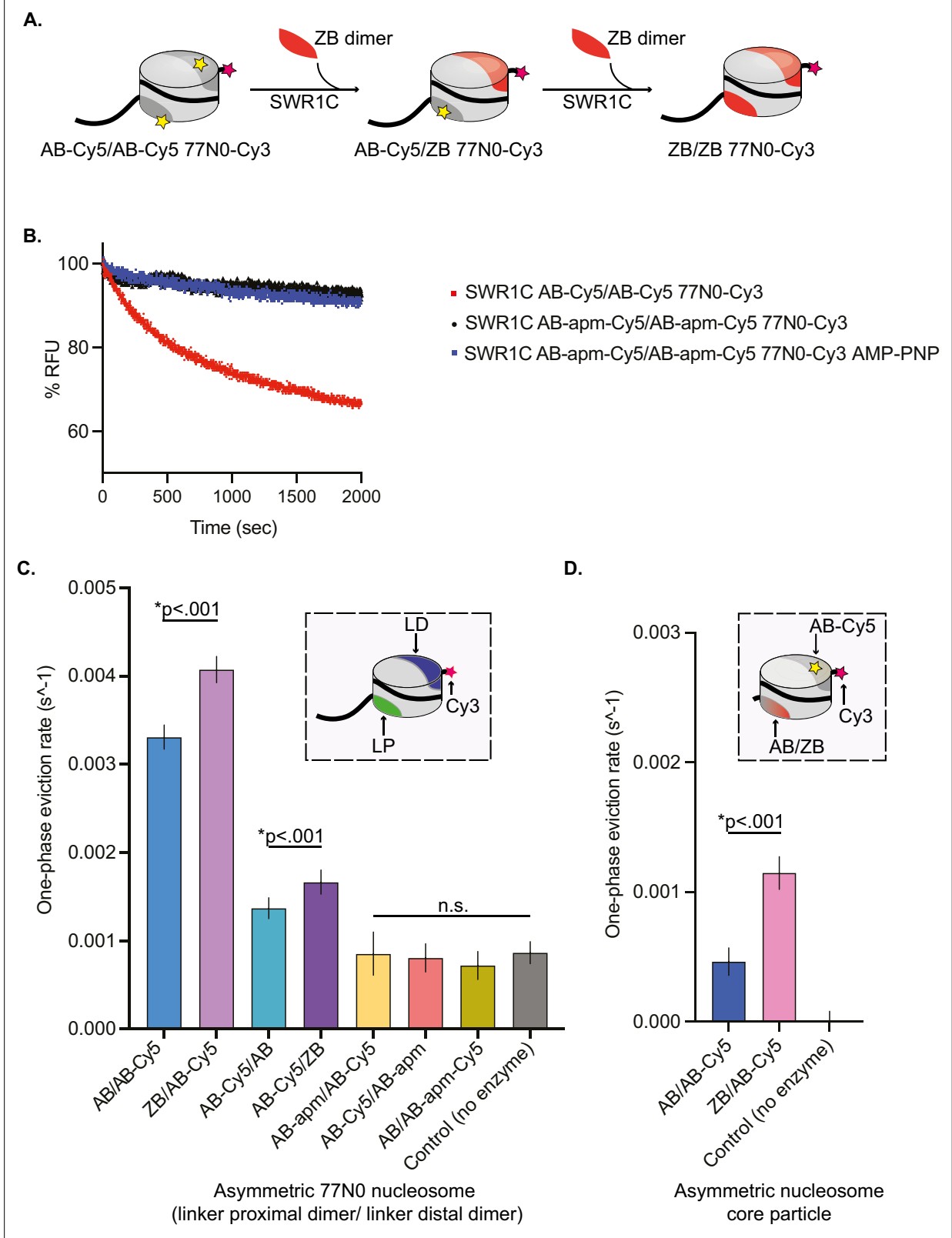

**Figure 1.** H2A.Z deposition requires each nucleosomal acidic patch. (**A**) Schematic of SWR1C-mediated dimer exchange. Cylinder is representative of a nucleosome; light gray area of cylinder represents H3/H4 tetramer, dark gray area of cylinder represents AB heterodimer, yellow star represents Cy5, pink star represents Cy3, and solid black line represents DNA. The nucleosome undergoes two rounds of SWR1C-mediated dimer eviction where ZB heterodimers (orange) replace AB heterodimers. (**B**) 77N0-Cy3 nucleosomes were remodeled by SWR1C under single turnover conditions (30 nM

*Figure 1 continued on next page*

*Figure 1 continued*

SWR1C, 10 nM 77N0-Cy3 nucleosomes, 70 nM ZB dimers, 1 mM ATP or AMP-PNP). Dimer eviction is monitored by measuring Cy5 emission at 670 nm. SWR1C was able to perform dimer exchange on an AB-Cy5/AB-Cy5 nucleosome (red line) but was unable to exchange dimers on an AB-apm-Cy5/AB-apm-Cy5 nucleosome (black line). The AMP-PNP reaction contained the AB-apm-Cy5/AB-apm-Cy5 nucleosomal substrate. (**C**) Asymmetrically assembled 77N0-Cy3 and Cy3-0N77 nucleosomes were remodeled by SWR1C under single turnover conditions (50 nM SWR1C, 10 nM nucleosomes, 50 nM ZB dimers, 1 mM ATP) and one-phase eviction rates were calculated in Prism 9 and plotted. The dimer labels are listed in linker proximal (LP), linker distal (LD) order (see inset for template nucleosome represented as a cylinder with H3/H4 tetramer in light gray, LP dimer in green, LD dimer in dark blue, DNA as a solid black line with Cy3 (pink star) on the 0bp-linker side). Nucleosomes with a ZB dimer contralateral to the dimer being evicted had significantly faster rates for both linker distal (first two bars) and linker proximal (second two bars). Linker distal eviction was also significantly faster than linker proximal eviction. Rates for nucleosomes containing an AB-apm dimer contralateral to the dimer being evicted (fifth and sixth bars) were not significantly different from no enzyme controls (last bar) in either orientation. SWR1C was also unable to evict a dimer with an acidic patch mutant (seventh bar). (**D**) Asymmetrically assembled 0N0-Cy3 nucleosomes were remodeled by SWR1C under single turnover conditions (50 nM SWR1C, 10 nM nucleosomes, 50 nM ZB dimers, 1 mM ATP) and one-phase eviction rates were calculated in Prism 9 and plotted. Nucleosomes contained a single AB-Cy5 dimer and a contralateral AB or ZB dimer (see inset for template nucleosome represented as a cylinder with H3/H4 tetramer in light gray, AB-Cy5 dimer in dark gray, contralateral AB or ZB dimer represented by dark gray to orange gradient, 0N0-Cy3 DNA as a solid black line with Cy3 (pink star) label on one side). AB-Cy5/ZB 0N0-Cy3 nucleosomes had significantly faster eviction than AB-Cy5/AB 0N0-Cy3 nucleosomes. At least three independent nucleosome preparations were used for substrate, and error bars reflect 95% confidence intervals from at least three replicates.

The online version of this article includes the following source data and figure supplement(s) for figure 1:

**Source data 1.** Excel file containing raw data for experiments depicted in panels B–D of *Figure 1*.

**Figure supplement 1.** Gel-based assay for H2A.Z deposition.

**Figure supplement 1—source data 1.** Excel file containing raw data for experiments depicted in panel D of *Figure 1—figure supplement 1*.

**Figure supplement 1—source data 2.** Zip file containing raw and annotated images for gels shown in panels A–C of *Figure 1—figure supplement 1*.

**Figure supplement 2.** Strategy for assembly of asymmetric nucleosomes.

**Figure supplement 2—source data 1.** Zip file containing raw and annotated images for gels shown in panels B–D of *Figure 1—figure supplement 2*.

**Figure supplement 3.** Förster-resonance energy transfer (FRET)-based dimer eviction.

**Figure supplement 3—source data 1.** Excel file containing raw data for experiments depicted in panels A–I of *Figure 1—figure supplement 3*.

To further interrogate the role for each of the two nucleosomal acidic patches in dimer exchange by SWR1C, we performed dimer exchange reactions with asymmetrically assembled nucleosomal substrates. Such asymmetric nucleosomes allow the fluorescent labeling of only a single H2A/H2B dimer, such that we could monitor which dimer was targeted for replacement. We generated these substrates by leveraging the asymmetric features of the 601 nucleosome positioning sequence and the spontaneous incorporation of dimers into purified hexasomes (*Levendosky and Bowman, 2019*; *Figure 1—figure supplement 2A*). Hexasomes were generated by varying the ratio of *X. laevis* H3/H4 tetramers to yeast H2A/H2B heterodimers on 77N0 or 0N0 DNA that was labeled with Cy3 (*Figure 1—figure supplement 2B*). After hexasome purification, nucleosomes of different compositions were reconstituted by titration of the appropriate yeast heterodimer (*Figure 1—figure supplement 2C, D*). For nucleosomes with linker DNA, we list the asymmetric dimers in linker proximal, linker distal order throughout (e.g. AB/AB-apm for a nucleosome with a linker distal H2A/H2B dimer harboring the acidic patch alteration).

First, we monitored H2A.Z deposition with nucleosomes harboring an unlabeled H2A/H2B heterodimer with a disrupted acidic patch (AB-apm) assembled on the nucleosomal face opposite to the Cy5-labeled heterodimer that will be monitored for H2A.Z deposition (*Figure 1—figure supplement 3G and H*). Strikingly, H2A.Z deposition was crippled, irrespective of whether the linker distal or linker proximal H2A/H2B heterodimer was the target for exchange (*Figure 1C*, AB-apm/AB-Cy5 and AB-Cy5/AB-apm substrates). We then tested if the Cy5-labeled heterodimer that is targeted for replacement also requires an intact acidic patch (*Figure 1—figure supplement 3I*). In this case as well, H2A.Z deposition was not greater than the no enzyme control (*Figure 1C*, AB/AB-apm-Cy5 substrate). Thus, SWR1C requires that both faces of the nucleosome contain an intact acidic patch, irrespective of which H2A/H2B heterodimer is undergoing replacement.

Previously, we and others reported that SWR1C preferentially exchanges the linker distal H2A/H2B dimer, as monitored by either ensemble or single molecule FRET analyses (*Fan et al., 2022*; *Poyton et al., 2022*). The asymmetric nucleosome substrates, harboring only a single, labeled H2A/H2B dimer were leveraged to further investigate this preferential exchange. Consistent with previous results, a substrate with a labeled, linker distal heterodimer is exchanged at a ~threefold faster rate by SWR1C

compared to the linker proximal dimer (*Figure 1C*, compare AB/AB-Cy5 to AB-Cy5/AB; *Figure 1— figure supplement 3A, B*). To rule out the possibility that this preferential exchange was influenced by the unlabeled H2A/H2B dimer on the opposing nucleosomal face, nucleosomes harboring an unlabeled ZB dimer were also reconstituted. In these cases as well, the linker distal H2A/H2B dimer was exchanged at a more rapid rate, compared to the linker proximal heterodimer (*Figure 1C*, compare ZB/AB-Cy5 to AB-Cy5/ZB; *Figure 1—figure supplement 3D, E*). Surprisingly, incorporation of the ZB heterodimer on the contralateral (opposite to the dimer being evicted) nucleosome face stimulated H2A.Z deposition, irrespective of linker orientation (*Figure 1C*; purple bars). The stimulatory effect of contralateral ZB heterodimer placement remained even after elimination of the linker DNA (*Figure 1D*, *Figure 1—figure supplement 3C, F*). Interestingly, these assays also confirmed that linker DNA has a stimulatory impact on the H2A.Z deposition reaction, with both linker distal and linker proximal eviction occurring at a faster rate than eviction from a nucleosome core particle (*Ranjan et al., 2013*). To test whether the stimulation by linker DNA was due to differences in nucleosome-binding affinity, we ensured the experiment was done under saturating enzyme levels by repeating it at a fourfold increase in the concentration of SWR1C. At this higher concentration of SWR1C, the rates were unchanged, and the stimulatory impact of linker DNA remained (*Figure 2—figure supplement 1A*). Thus, both linker DNA and nucleosomal ZB dimers appear to stimulate H2A.Z deposition.

## Intact nucleosomal acidic patches are required for SWR1C ATPase activity

The ATPase activity of SWR1C is stimulated by both its nucleosomal substrate and free ZB dimers (*Luk et al., 2010*). Notably, the ATPase activity of SWR1C is not stimulated by a ZB/ZB nucleosome, implying that SWR1C can distinguish product from substrate (*Luk et al., 2010*). To investigate the role of the nucleosomal acidic patch in the stimulation of SWR1C ATPase activity, SWR1C ATPase activity was measured in the presence or absence of different 77N0-Cy3 nucleosomes. Consistent with previous results (*Luk et al., 2010*), SWR1C ATPase activity was stimulated by AB/AB nucleosomes as compared to the basal level. However, ATPase activity was not stimulated by either of the asymmetric acidic patch mutant nucleosomes (AB-apm/AB, AB/AB-apm) or by a nucleosome lacking both acidic patches (AB-apm/AB-apm), demonstrating the essential role of dual acidic patch engagement by SWR1C for stimulation of ATPase activity (*Figure 2—figure supplement 1B*).

## The nucleosomal acidic patch is a key driver of SWR1C-binding affinity

The inability of APM nucleosomes to stimulate the ATPase activity of SWR1C could be due to a defect in binding or the inability to form an active complex post-binding. To explore whether disruption of each acidic patch might impact the nucleosome-binding affinity of SWR1C for nucleosomes, we leveraged an FP assay. Using a Cy3-labeled 77N0 nucleosome, we determined a Kd of 13.6 nM for an AB/ AB nucleosome (*Figure 2A*), similar to prior studies (*Fan et al., 2022*). However, disruption of even a single nucleosomal acidic patch severely weakened SWR1C binding, such that a Kd was unable to be determined (*Figure 2B–D*). Given the range of SWR1C concentrations that could be used in this assay, we estimate that alteration of one or both nucleosomal acidic patches decreases binding affinity by at least 10-fold. These data indicate that the inability of APM nucleosomes to stimulate SWR1C ATPase activity is most likely due to a defect at the nucleosome-binding step ($K_m$), rather than $K_{cat}$, though an impact at this latter step cannot be eliminated as yet.

## Swc5 interacts with the nucleosomal acidic patch

Recently, a potential arginine anchor domain, the archetypal motif for proteins interacting with the nucleosomal acidic patch, was identified within the N-terminal, acidic domain of the SWR1C subunit, Swc2. However, removal of this domain (Swc2-ZN) has only a modest impact on the dimer exchange activity of SWR1C, and these data, combined with the essential nature of the nucleosomal acidic patches for SWR1C activity, suggest that additional subunits must contribute to acidic patch recognition (*Dai et al., 2021*). The Swc5 subunit is one candidate, as loss of Swc5 eliminates H2A.Z deposition and nucleosome-stimulated ATPase activity (*Wu et al., 2005*; see *Figure 3B, C*). Furthermore, removal of the Swc5 subunit from SWR1C crippled nucleosome binding, as measured by FP (*Figure 3D*). Previous studies of Swc5 identified an acidic N-terminal domain that interacts preferentially with AB dimers, and a conserved C-terminal domain, termed BCNT, that is essential for dimer exchange by

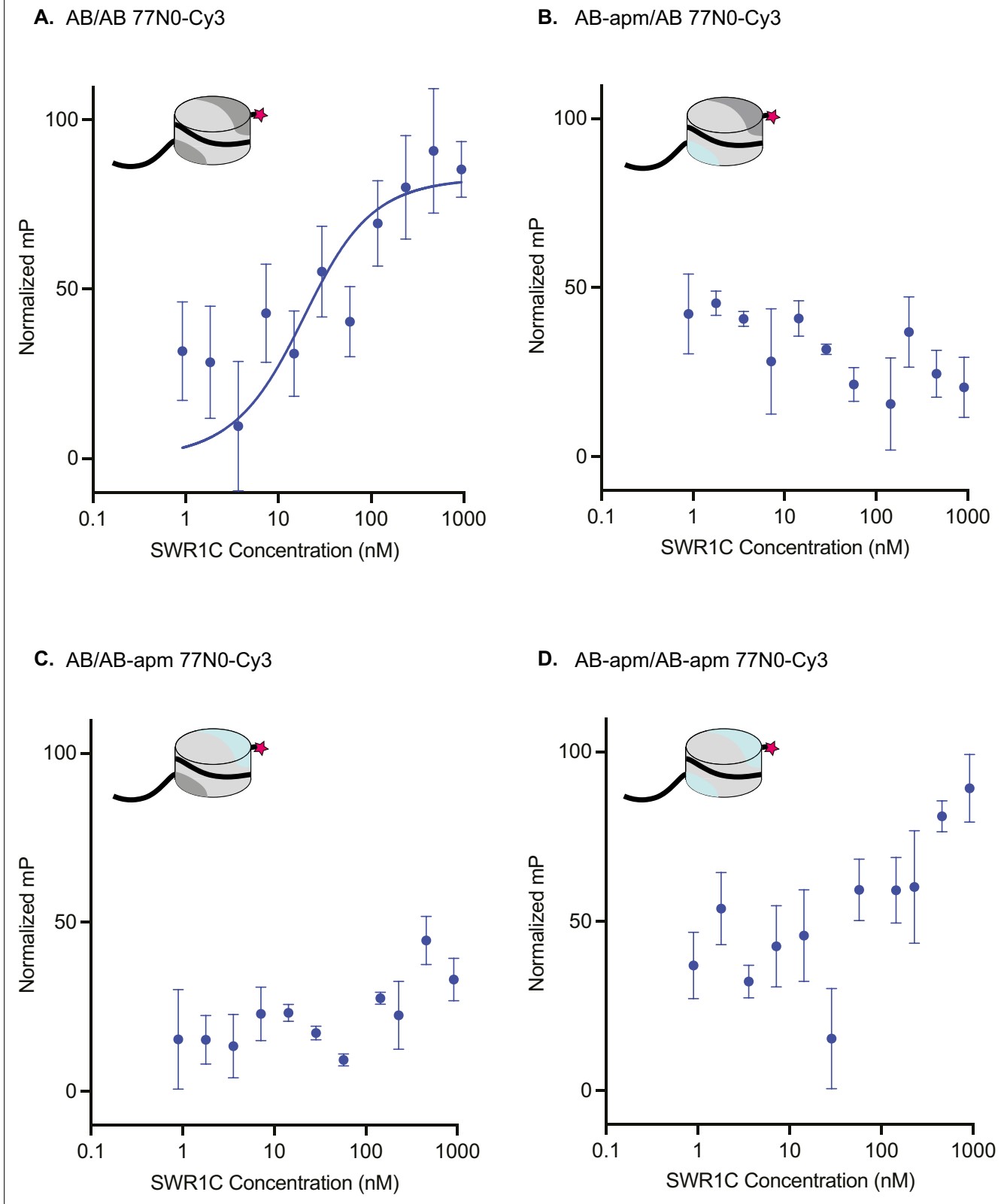

**Figure 2.** Nucleosomal acidic patches are required for SWR1C nucleosome binding. Asymmetrically assembled 77N0-Cy3 nucleosomes were incubated with SWR1C serially diluted in concentration from approximately 1 µM to 1 nM. Fluorescence polarization (FP) values were collected and plotted by concentration for each nucleosome type. Each graph has a representative nucleosome at the top left (represented by a cylinder with H3/H4 tetramer in light gray, AB dimer in dark gray, AB-apm dimer in light blue, 77N0-Cy3 DNA as a solid black line with Cy3 (pink star) label on the 0bp-linker side). FP

*Figure 2 continued on next page*

*Figure 2 continued*

data were normalized between experiments by translation, which does not effect the binding curve fit so that plots can be compared at a glance. (**A**) AB/AB 77N0-Cy3 nucleosomes showed a Kd as determined by a fit to the Morrison equation (blue line). The Kd calculated for each individual replicate was 13.55, 13.32, 13.94, and 13.75. The average of those individual values is 13.64. The standard deviation is 0.27. A Kd was unable to be calculated for nucleosomes with a linker proximal acidic patch mutant (**B**), a linker distal acidic patch mutant (**C**), or both acidic patches mutated (**D**) at the range of concentrations of SWR1C tested suggesting at least a 10-fold reduction in binding affinity. Error bars reflect 95% confidence intervals from at least three replicates.

The online version of this article includes the following source data and figure supplement(s) for figure 2:

**Source data 1.** Excel file containing raw data for experiments depicted in panels A–D of *Figure 2*.

**Figure supplement 1.** Dimer eviction and ATPase assays.

**Figure supplement 1—source data 1.** Excel file containing raw data for experiments depicted in panels A–C of *Figure 2—figure supplement 1*.

SWR1C (*Sun and Luk, 2017*; *Huang et al., 2020*). An alignment of Swc5 homologs revealed a region adjacent to the BCNT domain, containing multiple arginine residues in a conserved basic region (*Figure 3A*).

To investigate the role of this Swc5 arginine-rich domain, alanine substitutions were created in four basic residues (RRKR), and this derivative was recombinantly expressed and reconstituted into a SWR1C complex purified from a *swc5* deletion strain (SWR1C^swc5Δ^). In addition, SWR1C was reconstituted with wildtype Swc5, a Swc5 derivative lacking the acidic N-terminal domain (Swc5^79–303^), and a Swc5 derivative harboring alanine substitutions in the essential BCNT domain (Swc5^LDW-3A^). In all cases, derivatives were incorporated into SWR1C complexes with equal efficiencies (*Figure 3—figure supplement 1*). Consistent with previous studies, reconstitution with wildtype Swc5 fully restored dimer exchange activity to SWR1C^swc5Δ^ (*Figure 3B*, red curve). In contrast, SWR1C harboring Swc5^LDW-3A^ had no detectable exchange activity, and the complex that lacks the N-terminal, acidic domain of Swc5 (Swc5^79–303^) exhibited a modest defect, as expected from previous studies (*Figure 3B*; *Sun and Luk, 2017*). Strikingly, SWR1C that contained the Swc5^RRKR-4A^ derivative showed minimal activity in the FRET-based exchange assay (*Figure 3B*), and this complex exhibited only ~10% the activity of wildtype SWR1C in the gel-based H2A.Z deposition assay (*Figure 1—figure supplement 1A, C, D*). Dimer exchange activity for each SWR1C complex was also mirrored by their ATPase activity (*Figure 3D*). SWR1C ^swc5Δ^ lost nucleosome-stimulated ATPase activity, with or without free ZB dimers, as compared to wildtype. As expected, reconstitution of SWR1C ^swc5Δ^ with recombinant Swc5 rescued ATPase activity. Underscoring the importance of the arginine-rich domain, reconstitution of SWR1C ^swc5Δ^ with Swc5^RRKR-4A^ showed a large defect in nucleosome-stimulated ATPase activity, revealing its importance in both dimer exchange and ATPase activity. Additionally, we reinforced prior findings that the ATPase activity of the Swc5^LDW-3A^ complex was not stimulated by either nucleosomes or free ZB dimers, while the complex harboring Swc5^79–303^ had wildtype levels of ATPase activity (*Figure 2—figure supplement 1C*).

To further investigate whether the arginine-rich domain within Swc5 binds to the nucleosomal acidic patch, in vitro binding assays were performed with nucleosome core particles (0N0) and recombinant Swc5 (*Figure 4A–F*). Increasing amounts of Swc5 were incubated with nucleosomes, and binding was visualized by native-PAGE. Addition of wildtype Swc5 to nucleosomes led to formation of discrete complexes with an apparent Kd of 125.7 nM (*Figure 4A*). Likewise, binding of Swc5^LDW-3A^, which has a disrupted BCNT domain, bound to nucleosomes with nearly identical affinity, compared to wild-type Swc5 (*Figure 4B*). Importantly, high affinity binding of Swc5 to nucleosomes required an intact acidic patch, with binding reduced ~threefold when an APM nucleosome was used as the substrate (*Figure 4C*). Furthermore, the Swc5^RRKR-4A^ derivative also showed an ~threefold loss in affinity, with an apparent Kd of 379.1 nM (*Figure 4D*). Together these data demonstrate that Swc5 has nucleosome-binding activity, and the data are consistent with direct interactions between the Swc5 arginine-rich domain and the nucleosomal acidic patch.

A time-resolved FRET assay (TR-FRET) was employed (*Figure 5A*) as an additional solution approach to monitor the binding of Swc5 to nucleosomes (*Wesley et al., 2022*). In this assay, increasing concentrations of recombinant, 6His-tagged Swc5 were incubated with a 31N1 nucleosome. The 6His-Swc5 protein were labeled with a ULight alpha-6xHIS acceptor antibody, and biotinylated nucleosomal DNA was labeled with a Eu-streptavidin donor fluorophore (*Figure 5A*). The binding of Swc5 to the nucleosome was monitored by quantitative measurements of the time-resolved FRET signal between

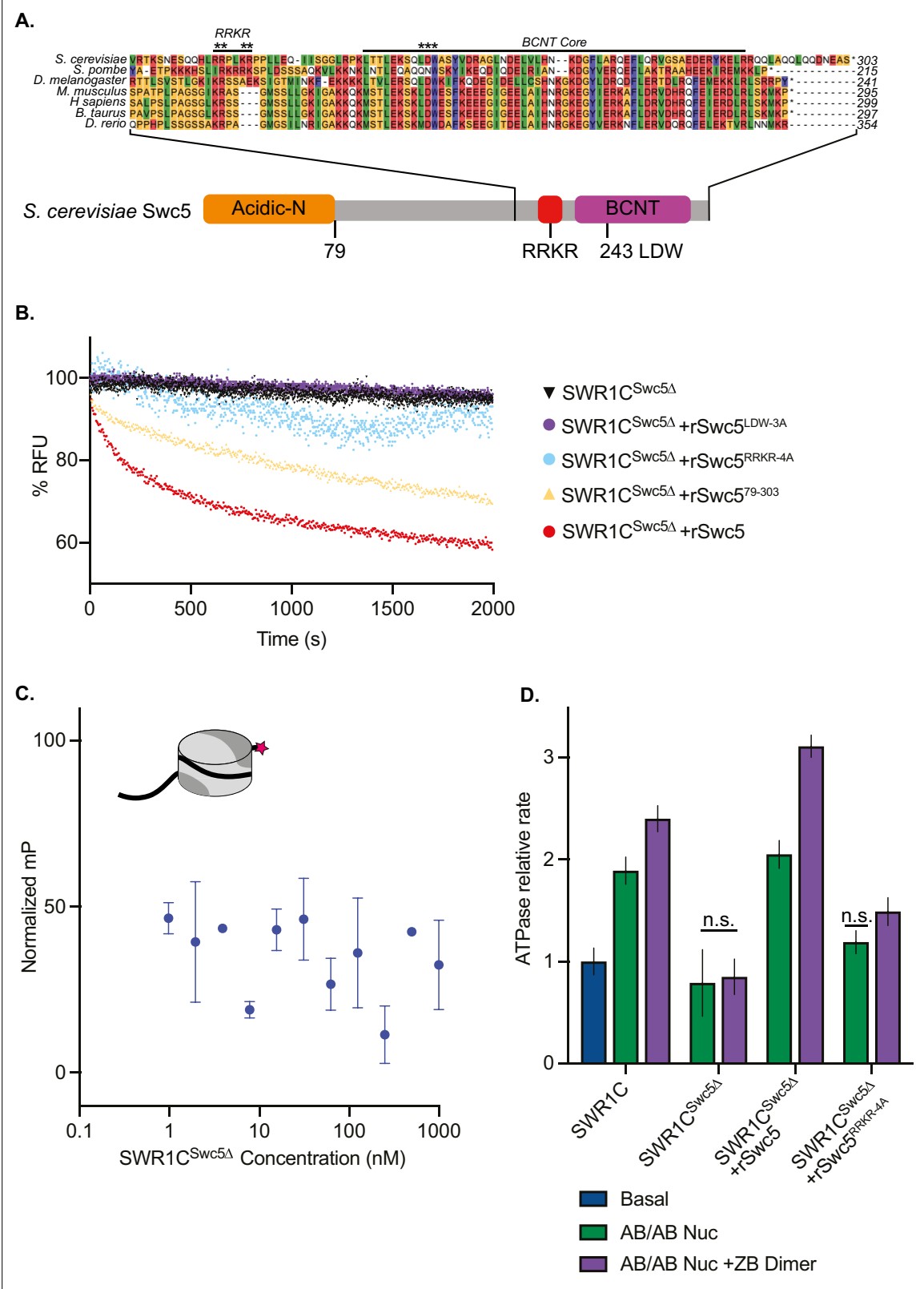

**Figure 3.** Swc5 contains a key arginine-rich motif. (**A**) Alignment of SWR1C subunit Swc5 homologs reveals a conserved arginine in a region we named the RRKR motif in *S. cerevisiae* adjacent to the essential BCNT region and conserved LDW residues (asterisks are used to denote residues mutated in RRKR-4A and LDW-3A mutants). A linear schematic of this gene with the acidic N-terminal domain in orange, gene body in gray, RRKR motif in red, and BCNT region in purple shows relative locations of these domains. (**B**) Swc5 and Swc5 derivatives were reconstituted into SWR1C$^{Swc5\Delta}$

*Figure 3 continued on next page*

*Figure 3 continued*

complexes to measure their activity on dimer exchange using Förster-resonance energy transfer (FRET) at single turnover conditions (30 nM SWR1C, 10 nM 77N0-Cy3 AB-Cy5/AB-Cy5 nucleosomes, 70 nM ZB dimers, 1 mM ATP). The relative fluorescence unit (RFU) for each reaction was normalized and plotted as a function of time. SWR1C$^{Swc5\Delta}$ was unable to carry out dimer exchange (black triangles), and recombinant Swc5 (rSwc5) rescued activity (red circles). SWR1C$^{Swc5\Delta}$ reconstituted with Swc5 containing mutations in the LDW (rSwc5$^{LDW-3A}$) or RRKR motifs (rSwc5$^{RRKR-4A}$) showed no dimer eviction activity (purple and blue circles, respectively), while an N-terminal truncation of Swc5 (rSwc5$^{79-303}$) had a reduction in activity compared to full-length Swc5 (yellow triangles). (**C**) Asymmetrically assembled 77N0-Cy3 nucleosomes were incubated with SWR1C$^{Swc5\Delta}$ serially diluted in concentration from approximately 1 μM to 1 nM. Fluorescence polarization values were collected and plotted by concentration. A representative nucleosome is shown at the top left (represented by a cylinder with H3/H4 tetramer in light gray, AB dimer in dark gray, 77N0-Cy3 DNA as a solid black line with Cy3 (pink star) label on the 0bp-linker side). (**D**) Nucleosomal stimulation of SWR1C ATPase activity with (purple bars) or without (green bars) ZB dimers added was measured for complexes containing SWR1C, SWR1C$^{Swc5\Delta}$, SWR1C$^{Swc5\Delta}$+rSwc5, and SWR1C$^{Swc5\Delta}$+rSwc5$^{RRKR-4A}$ using a phosphate sensor assay. Calculated rates were normalized to basal SWR1C activity (blue bar). Stimulation of ATPase activity was lost in the SWR1C$^{Swc5\Delta}$ complex and reduced in the SWR1C$^{Swc5\Delta}$+rSwc5$^{RRKR-4A}$ complex. Error bars reflect 95% confidence intervals from at least three replicates.

The online version of this article includes the following source data and figure supplement(s) for figure 3:

**Source data 1.** Excel file containing raw data for experiments depicted in panels B, C, and D of *Figure 3*.

**Figure supplement 1.** Reconstitution of SWR1C with Swc5 derivatives.

**Figure supplement 1—source data 1.** Zip file containing raw and annotated images for gels (numbered 1 through 4 from left to right) shown in *Figure 3—figure supplement 1*.

---

the ULight alpha acceptor and the Eu fluorophore. Binding curves for wildtype Swc5 yielded an apparent Kd of 133 ± 12 nM, a value that agrees well with the Kd measured by the nucleosome gel shift assay. Binding experiments performed in parallel with the Swc5$^{RRKR-4A}$ derivative yielded a Kd of 593 ± 28 nM, confirming that the arginine-rich motif within Swc5 plays a key role in nucleosome recognition (*Figure 5B*).

While the gel shift and TR-FRET assays establish that Swc5 can bind to nucleosomes, data from these assays can only indirectly link the nucleosomal acidic patch to the Swc5 arginine-rich domain. To probe for more direct interactions, a fluorescence quenching assay was employed. These assays exploit nucleosomes harboring a site-specific, Oregon Green fluorophore whose emission is sensitive to the chemical environment whereby fluorescence is quenched by protein binding (*McGinty et al., 2014*; *Winkler et al., 2012*). Nucleosome core particles were reconstituted that contained an Oregon Green fluorophore covalently attached to either the histone H4 N-terminal domain (H4-tail) (*Figure 6A*, top) or to a residue directly adjacent to the nucleosomal acidic patch (*Figure 6A*, bottom). Titration of wildtype Swc5 led to the concentration-dependent quenching of the Oregon Green fluorophore positioned at the acidic patch, but little quenching was observed for the fluorophore located on the H4-tail (*Figure 6B*) demonstrating the sensitivity of this assay. Importantly, Swc5 did not quench the acidic patch probe when this substrate also harbored alanine substitutions within the acidic patch (*Figure 6B*). Examined in the same titration of concentrations, we found that the Swc5$^{LDW-3A}$ derivative also showed specific quenching of the acidic patch probe (*Figure 6C*), but the Swc5$^{RRKR-4A}$ derivative was defective for quenching the acidic patch probe (*Figure 6D*). Together these results are consistent with a direct interaction between the arginine-rich domain of Swc5 and the nucleosomal acidic patch, an interaction that is essential for dimer exchange by SWR1C.

## Structure of a Swc5–nucleosome complex

We analyzed the structure of Swc5$^{79-303}$ in complex with a 147-bp nucleosome core particle using cryo-EM (*Figure 7*). We collected over 13,000 micrographs of the cross-linked sample and were able to easily resolve the nucleosome structure to 3 Å resolution (*Figure 7—figure supplement 1L–O*). On the other hand, the moiety corresponding to Swc5 displayed severe structural heterogeneity, hindering our ability to build an atomic model for Swc5. The protein appears to be flexible on the nucleosome, with continuous conformational dynamics apparent (*Figure 7A–D*, *Figure 7—video 1*). Nevertheless, reconstructions of particle subsets obtained by local 3D classification show that Swc5 forms multivalent interactions with the nucleosome via the nucleosome acidic patch, the H4 N-terminal histone tail, and the C-terminal helix of H2B (*Figure 7E–G*). In the majority of 3D reconstructions, we observe additional density at the nucleosome acidic patch attributable to an arginine residue, like in other complexes with arginine anchor interactions (*Makde et al., 2010*; *McGinty and Tan, 2015*). Unfortunately, the low local resolution precludes the determination of the precise Swc5

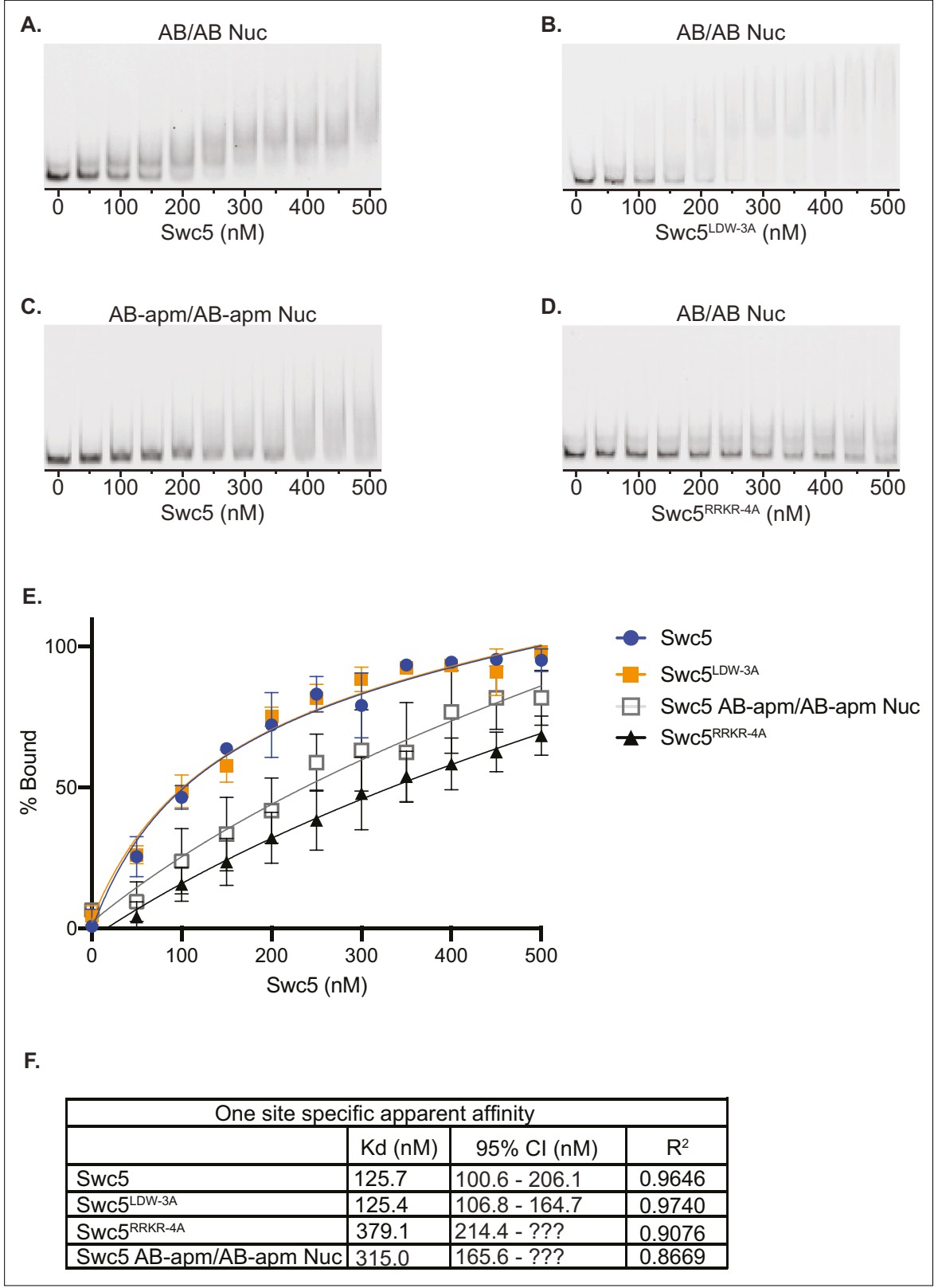

| One site specific apparent affinity | | | |
|---|---|---|---|
| | Kd (nM) | 95% CI (nM) | R² |
| Swc5 | 125.7 | 100.6 - 206.1 | 0.9646 |
| Swc5^LDW-3A | 125.4 | 106.8 - 164.7 | 0.9740 |
| Swc5^RRKR-4A | 379.1 | 214.4 - ??? | 0.9076 |
| Swc5 AB-apm/AB-apm Nuc | 315.0 | 165.6 - ??? | 0.8669 |

**Figure 4.** Swc5 is a nucleosome-binding subunit. Gel mobility shift assays were performed with 5 nM 0N0 nucleosomes and recombinant Swc5 ranging in concentration from 0 to 500 nM, in 50 nM increments. Swc5 was able to bind AB/AB nucleosomes (**A**) but showed reduced binding of AB-apm/AB-apm nucleosomes (**B**). Swc5^LDW-3A bound AB/AB nucleosomes (**C**) whereas Swc5^RRKR-4A showed reduced binding (**D**). Percent of nucleosomes bound for

*Figure 4 continued on next page*

*Figure 4 continued*

each condition was calculated using ImageQuant and plotted by Swc5 concentration (**E**) and specific binding Kd values were predicted along with 95% confidence intervals (CIs) predicted in Prism 9 (**F**).

The online version of this article includes the following source data for figure 4:

**Source data 1.** Excel file containing raw data for experiments depicted in panel E of *Figure 4*.

**Source data 2.** Zip file containing raw and annotated images for gels shown in panels A–D of *Figure 4*.

arginine residue involved in this interaction (*Figure 7—figure supplement 1D–G*). In contrast, the 3 Å resolution of the nucleosome core allows the identification of H2B-K123 as an interacting residue with Swc5. In most 3D classes, Swc5 forms either a partial or a complete bridge between H2B C-terminal helix including K123 and the N-terminal tail of H4.

## The Swc5 arginine-rich domain is required for SWR1C function in vivo

Previous studies have shown that yeast lacking functional SWR1C grow poorly on media containing formamide (*Kobor et al., 2004*; *Sun and Luk, 2017*). To investigate the functional role of the Swc5 arginine-rich domain in vivo, growth assays were performed with isogenic *swc5Δ* strains that harbor low copy vectors that express different Swc5 derivatives (*Figure 8*). As expected, the *swc5Δ* strain grew well on rich media, but was impaired on formamide media (vector), while cells expressing wild-type Swc5 grew well on both media (*SWC5*). Consistent with a previous study, a strain expressing the derivative with a disrupted BCNT domain showed slow growth on formamide media (Swc5[LDW-3A]), and the Swc5[79–303] derivative that lacks the acidic N-terminal domain showed a moderate growth defect (*Sun and Luk, 2017*). Importantly, a strain expressing the Swc5[RRKR-4A] derivative had a severe growth defect on formamide media, consistent with an important role of the arginine-rich region for SWR1C function.

## Discussion

The nucleosomal acidic patch has emerged as a key binding pocket for nearly all ATP-dependent chromatin remodeling complexes. Here, leveraging asymmetrically constructed nucleosomes, we have

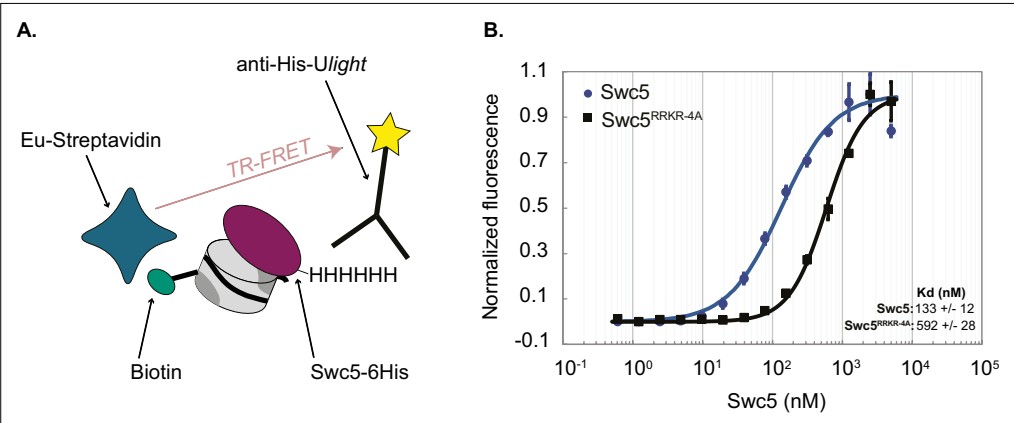

**Figure 5.** Time-resolved Förster-resonance energy transfer (TR-FRET) assay confirms Swc5 nucleosome-binding activity. (**A**) Schematic of the TR-FRET assay. Recombinant 6His-Swc5 is labeled with a ULight alpha 6His acceptor antibody, and biotinylated nucleosomal DNA is labeled with an Eu-streptavidin acceptor fluorophore. The Förster-resonance energy transfer (FRET) signal increases as 6His-Swc5 binds to the nucleosome. (**B**) TR-FRET assay was performed with 2 nM 31N1 nucleosomes and recombinant Swc5 or Swc5[RRKR-4A] ranging in concentration from 0.6 to 6000 nM. Recombinant wildtype Swc5 bound with an apparent Kd of ~133 nM, while the Swc5[RRKR-4A] derivative bound with an apparent Kd of ~592 nM. Note that the Swc5[RRKR-4A] assays did not reach full saturation, and thus the measured Kd is likely an under-estimate of the reported value. Kd values were determined from triplicate titrations of each Swc5 variant and are reported as means ± standard error of the mean.

The online version of this article includes the following source data for figure 5:

**Source data 1.** Excel file containing raw data for experiments depicted in panel B of *Figure 5*.

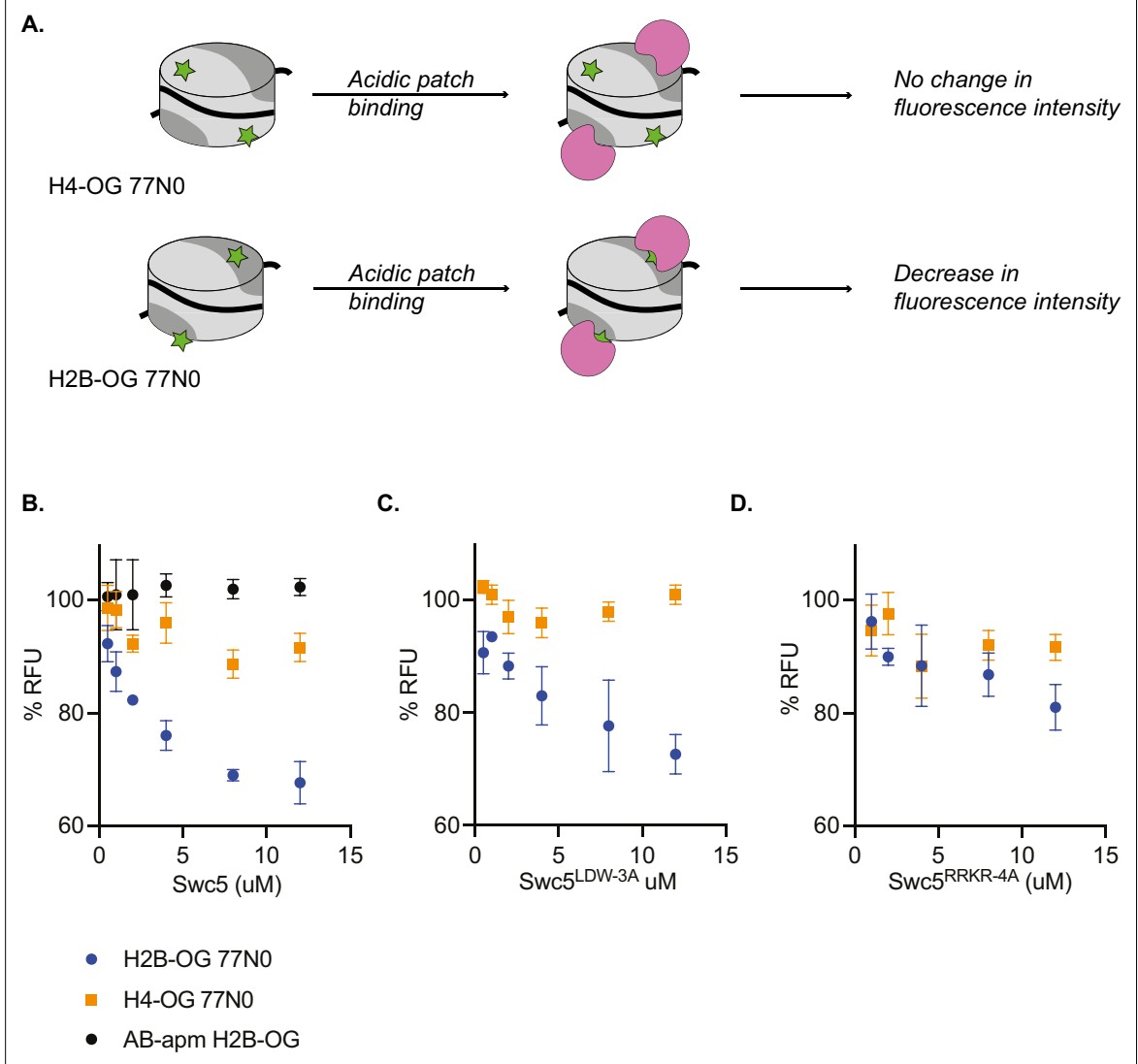

**Figure 6.** Swc5 interacts with the nucleosome acidic patch. (**A**) Schematic of Oregon green based fluorescence quenching assay. Top shows a nucleosome (represented as a cylinder with H3/H4 tetramer in light gray, AB dimers in dark gray, and 0N0 DNA as a solid black line) with an Oregon green label (green star) on H4. Oregon green is exquisitely sensitive to changes in its local environment. Binding of an acidic patch interacting protein (pink) does not interact with Oregon green label on H4, resulting in no change in fluorescence intensity. A label on H2B near the acidic patch (bottom) is affected by an acidic patch interacting protein, resulting in decreased fluorescence intensity. (**B**) Swc5 was added to 10 nM H4-OG AB/AB 0N0 nucleosomes (orange squares), H2B-OG AB/AB 0N0 nucleosomes (blue circles), and H2B-OG AB-apm/AB-apm 0N0 nucleosomes (black circles) resulting in quenching of the AB/AB H2B-OG 0N0 nucleosomes but not the H4-OG AB/AB 0N0 nucleosomes or the H2B-OG AB-apm/AB-apm 0N0 nucleosomes. (**C**) Swc5[LDW-3A] similarly quenched H2B-OG AB/AB 0N0 nucleosomes with no change in H4-OG AB/AB 0N0 nucleosomes while Swc5[RRKR-4A] (**D**) did not show significant quenching in either condition. Error bars reflect standard deviations from at least three replicates.

The online version of this article includes the following source data for figure 6:

**Source data 1.** Excel file containing raw data for experiments depicted in panels B–D of *Figure 6*.

found that both acidic patches are essential for the ATP-dependent deposition of H2A.Z by SWR1C. Each acidic patch is key for high affinity nucleosome binding, and this defect in binding appears to result in an inability of APM nucleosomes to stimulate SWR1C ATPase activity. The requirement for both acidic patches suggests that SWR1C adopts a pincer conformation, contacting both acidic patches and leading to a conformation that stimulates the ATPase activity of SWR1C from basal levels. Like others have noted (*Clapier, 2021*), the thermodynamic landscape of the nucleosome provides a much higher barrier to dimer exchange than sliding, and we suspect this is likely responsible for the greater effect of an asymmetric acidic patch mutant nucleosome on SWR1C as compared to other

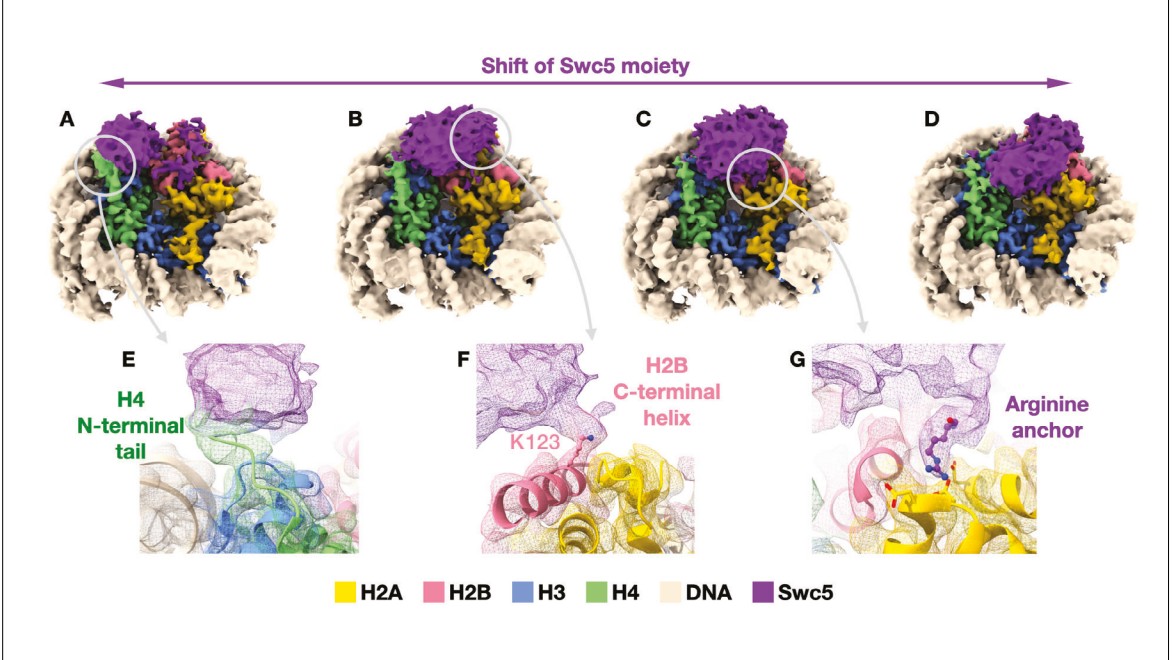

**Figure 7.** Cryo-EM structure of Swc5/nucleosome complex. (**A–D**) Reconstructions of selected particle subsets obtained from 3D classification that was focused on the Swc5 moiety (purple) illustrate the dynamic nature of Swc5 binding to the nucleosome. Three interactions persist among the 3D classes: (**E**) Swc5 interaction with the N-terminal tail of histone H4 (green), (**F**) Swc5 interaction with the C-terminal helix of histone H2B including K123, and (**G**) an Arginine anchor bound into the acidic patch of the histone core. The gray circles and arrows denote the reconstructions and locations from where the cut-out panels originate.

The online version of this article includes the following video and figure supplement(s) for figure 7:

**Figure supplement 1.** Cryo-EM analysis of Swc5/nucleosome complex.

**Figure 7—video 1.** Conformational heterogeneity of Swc5.

https://elifesciences.org/articles/94869/figures#fig7video1

remodelers. Additionally, while the acidic patch plays a key role in the activity of SWR1C and other chromatin remodeling enzymes, it also serves as a docking point for other non-chromatin remodeling nuclear proteins (*McGinty and Tan, 2021*; *Kalashnikova et al., 2013*; *Skrajna et al., 2020*). From our data, we can conclude that other acidic patch interacting proteins may consequently inhibit the activity of SWR1C even if only one acidic patch is sterically occluded.

Recently, the nucleosomal acidic patch has been shown to differentially impact the behavior of chromatin remodeling enzymes in an asymmetric manner (*Levendosky et al., 2016*; *Lehmann et al., 2020*; *Levendosky and Bowman, 2019*). Studies on the sliding activity of the ISWI chromatin remodeling enzyme from both humans and *Drosophila melanogaster* show that the linker proximal acidic patch (the DNA entry side) is more important than the linker distal acidic patch for maintaining appropriate sliding activity (*Levendosky and Bowman, 2019*). In contrast, budding yeast Chd1 shows a small reduction in sliding rate when either acidic patch is disrupted. The nucleosomal acidic patch is also essential for remodeling by SWI/SNF family members, and in this case the two acidic patches on the nucleosome are contacted by two different subunits –Snf5 (human SMARCAD1) and the Snf2 (human BRG1) ATPase (*Han et al., 2020*; *Wagner et al., 2020*; *Ye et al., 2019*; *Mashtalir et al., 2020*). Cramer and colleagues suggested that the RSC complex may use both acidic patches to sandwich the histone octamer (*Wagner et al., 2020*), however, loss of the Snf5 (SMARCAD1) arginine anchor has only a small impact on remodeling, whereas the key domain within the Snf2 subunit (the SnAC domain) is essential for activity (*Ye et al., 2019*; *Sen et al., 2011*; *Mulvihill et al., 2021*). Thus, unlike SWR1C, both acidic patch surfaces are not essential for the activity of other remodeling enzymes.

The cryoEM structure of a SWR1C–nucleosome complex suggests that the enzyme has two major nucleosomal contacts – the Swr1 ATPase lobes interact with nucleosomal DNA ~2 helical turns from the nucleosomal dyad (SHL2.0), and the Swc6/Arp6 module interacts with DNA at the nucleosomal

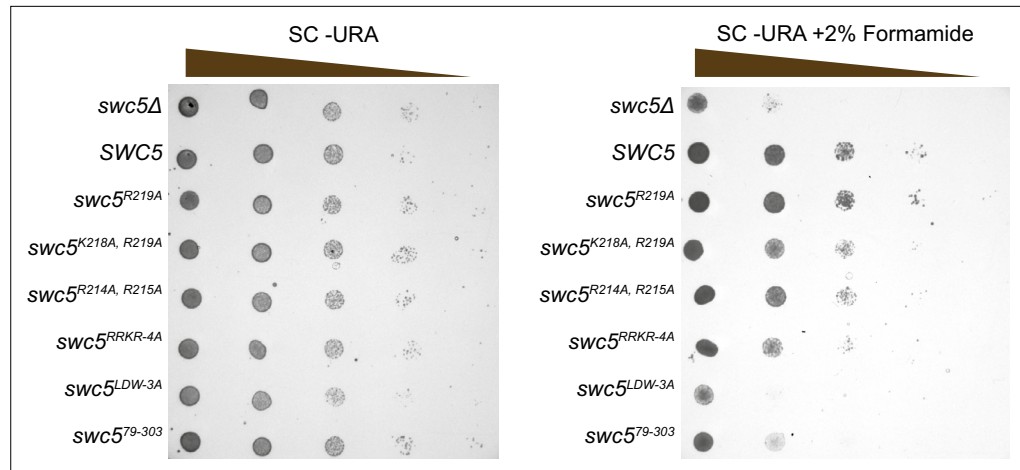

**Figure 8.** Swc5 arginine-rich region is key for SWR1C activity in vivo. Genetic complementation using a *swc5Δ* strain (*MATa his3Δ1 leu2Δ0 met15Δ0 ura3Δ0 swc5Δ::kanMX4*) transformed with an empty pRS416 CEN/ARS *URA3* vector (top row in both panels) or the same vector with various alleles of *SWC5*. Cultures were grown to an OD of 1 and spotted with serial dilution onto plates with synthetic complete media lacking uracil without (left) or with (right) 2% formamide.

The online version of this article includes the following source data for figure 8:

**Source data 1.** Zip file containing raw and annotated images for gels (referred to as left and right panel) shown in *Figure 8*.

edge (*Willhoft et al., 2018*). These contacts encompass the DNA gyre that wraps the AB dimer that is destined to be exchanged in the first round of catalysis (the linker distal dimer; *Singh et al., 2019*). The structure also indicated that each of the two nucleosomal acidic patches may make contacts with SWR1C subunits, although in both cases the unambiguous identification of the amino acid sequence was restricted by the resolution of cryoEM maps. It seems likely that the Swc2 subunit may contact the acidic patch of the H2A/H2B dimer that is not targeted for replacement. Unfortunately, the Swc5 subunit was not visualized in a previous cryoEM structure (*Willhoft et al., 2018*), but here we find that Swc5 can make an arginine anchor-like interaction with the nucleosomal acidic patch. SWR1C that lacks Swc5 shows the same spectrum of defects as disruption of the nucleosome acidic patch, consistent with Swc5 playing a key role in anchoring SWR1C to the nucleosome. Based on the existing cryoEM map of the SWR1C–nucleosome complex, we propose that Swc5 contacts the H2A/H2B dimer that is targeted for eviction, positioned between the Arp6/Swc6 complex and the Swr1 ATPase. While some observed density at that nucleosome acidic patch was tentatively identified as Swc6 in the cryoEM map, the authors noted that the resolution in this region was insufficient to make a putative sequence determination (*Willhoft et al., 2018*). Notably, this location places Swc5 close to its interaction surfaces on Swr1 (*Lin et al., 2017*; *Wu et al., 2009*), as well as positioning its acidic N-terminal domain in an ideal location to capture the evicted AB dimer, as previously suggested (*Huang et al., 2020*).

One advantage of the hexosome strategy for reconstitution of asymmetric nucleosomal substrates is that we were able to reconstitute homogeneous, heterotypic Z/A nucleosomes. This nucleosome mimics the product of the first round of H2A.Z deposition, a potential intermediate toward the formation of a homotypic Z/Z nucleosome. Surprisingly, we found that incorporation of one H2A.Z/H2B dimer stimulated the second round of dimer exchange. This stimulation was observed for both linker proximal and linker distal positions, as well as for a nucleosome core particle. The heterotypic Z/A nucleosome (with linker) reproducibly stimulated deposition rate by ~20%, values which are unlikely to be detected in prior studies that used a standard A/A nucleosomal substrate and reaction conditions where nucleosomes were in large excess (*Luk et al., 2010*). How incorporation of one H2A.Z/H2B dimer stimulates the second cycle of dimer exchange in not clear, but one possibility is that re-orientation or re-binding of SWR1C may be facilitated if the enzyme encounters an H2A.Z surface on the initial binding event.

Previous studies have investigated the impact of Swc5 on the nucleosome-binding activity of SWR1C, concluding that this subunit does not play a role in nucleosome recognition. For instance, initial work from Wu and colleagues employed a nucleosomal array pulldown assay (*Ranjan et al., 2013*), while subsequent studies from Wigley and colleagues used a native gel, electrophoresis shift assay (*Lin et al., 2017*). In both cases, loss of Swc5 from SWR1C had no detectable impact on nucleosome binding. Furthermore, recent live cell imaging from the Wu group showed that Swc5 is not required for general chromatin association by SWR1C (*Ranjan et al., 2020*). Together with our results, these data suggest that SWR1C may make strong nonspecific interactions with chromatin, most likely through the Swc2 subunit which is known to bind DNA (*Ranjan et al., 2013*). Indeed, deletion of Swc2 results in a substantial reduction in the localization of SWR1C to chromatin in vivo (*Ranjan et al., 2020*). Furthermore, stabilization of a SWR1C–nucleosome complex in a gel matrix may mask the defects in binding that we observe here in a solution-based assay. Notably, our FP assay is unable to detect nucleosome binding by SWR1C if the affinity is reduced by more than 10-fold, and consequently our work indicates that Swc5 is essential for high affinity interactions that are required for subsequent H2A.Z deposition. Whether Swc5 plays roles subsequent to nucleosome recognition is currently unclear.

In addition to contacting the nucleosomal acidic patch, our Swc5–nucleosome structure indicates that Swc5 makes contacts with the H4 N-terminal tail and the H2B C-terminal helix. Previous work has shown that acetylation of the H4 tail can influence both SWR1C recruitment and activity, raising the possibility that the Swc5–H4 interaction may be functionally important (*Altaf et al., 2010*). Interestingly, we find that Swc5 interacts with the H2B C-terminal helix via contact with H2B-K123 which is subject to transcription-associated modification with ubiquitin by the Bre1-Rad6 ubiquitin ligase complex. Furthermore, H2B-K123ub is enriched at the +1 nucleosome, the same nucleosome targeted by SWR1C for H2A.Z deposition. Strikingly, a ChIP-exo study found that H2B-K123ub is enriched at the NFR proximal face of the +1, while H2A.Z is enriched at the NFR distal surface (*Rhee et al., 2014*). The anti-correlation between H2B-K123ub and H2A.Z is consistent with the idea that H2B ubiquitinylation disrupts the Swc5–nucleosome interaction and regulates H2A.Z deposition. Interestingly, H2B-K123ub is also known to regulate the Chd1 remodeler, but in this case H2B-K123ub does not appear to contact Chd1, but rather it interacts with unwrapped nucleosomal DNA, stimulating sliding activity (*Sundaramoorthy et al., 2018*).

The human homolog of Swc5 is Craniofacial development protein 1 (Cfdp1), although there is currently a lack of information on the role of Cfdp1 in development. Mutations in a human homolog of SWR1C, SRCAP is implicated in Floating Harbor syndrome, which results in skeletal and facial defects (*Messina et al., 2016*). Complementary to this observation, additional evidence suggests Cfdp1 has an essential role in bone development (*Celauro et al., 2017*). Interestingly, a survey of Cfdp1 mutations identified in the TCGA transCancer atlas reveals many missense substitutions within the C-terminal BCNT domain, as well as a cluster of substitutions directly adjacent to the arginine-rich domain. The majority of these Cfdp1 alterations are associated with uterine endometrial carcinomas, suggesting that Cfdp1, and likely the SRCAP remodeler, are key for suppressing cancer initiation or progression in this cell type.

# Materials and methods

## Key resources table

| Reagent type (species) or resource | Designation | Source or reference | Identifiers | Additional information |
| --- | --- | --- | --- | --- |
| Strain, strain background (*S. cerevisiae*) | *W1588-4C swr1::SWR1–3xFLAG-P-kanMx-P htz1Δ::natMX* | Ed Luk (SUNY Stony Brook) | yEL190 | |
| Strain, strain background (*S. cerevisiae*) | *W1588-4C swr1::SWR1–3xFLAG-P-kanMx-P htz1Δ::natMX swc5Δ::hphMX* | This paper | CY2535 | Available from Peterson lab |

*Continued on next page*

*Continued*

| Reagent type (species) or resource | Designation | Source or reference | Identifiers | Additional information |
|---|---|---|---|---|
| Strain, strain background (*S. cerevisiae*) | *MATa his3Δ1 leu2Δ0 met15Δ0 ura3Δ0 YBR231c::kanMX4* | Euroscarf | Y03371 | |
| Strain, strain background (*E. coli*) | Rosetta 2 (DE3) | Novagen | 70954 | |
| Strain, strain background (*E. coli*) | DH5alpha | Thermo Fisher Scientific | EC0112 | |
| Antibody | ULight alpha-6xHIS acceptor antibody | Perkin Elmer | TRF0134-D | Mouse monoclonal anti poly Histidine tag used at 1:20 molar ratio to His-tagged Swc5 variants |
| Recombinant DNA reagent | pRS416 (plasmid) | ATCC | 87521 | CEN/ARS/URA3 vector |
| Recombinant DNA reagent | CP1566 (plasmid) | This paper | | *SWC5* version of pRS416 Available from Peterson lab |
| Recombinant DNA reagent | CP1567 (plasmid) | This paper | | *swc5 R219A* version of pRS416 Available from Peterson lab |
| Recombinant DNA reagent | CP1568 (plasmid) | This paper | | *swc5 R214A, R215A* version of pRS416 Available from Peterson lab |
| Recombinant DNA reagent | CP1569 (plasmid) | This paper | | *swc5 K218A, R219A* version of pRS416 Available from Peterson lab |
| Recombinant DNA reagent | CP1579 (plasmid) | This paper | | *swc5 RRKR-4A* version of pRS416 Available from Peterson lab |
| Recombinant DNA reagent | CP1580 (plasmid) | This paper | | *swc5 LDW-3A* version of pRS416 Available from Peterson lab |
| Recombinant DNA reagent | CP1581 (plasmid) | This paper | | *swc5 79–303* version of pRS416 Available from Peterson lab |
| Recombinant DNA reagent | pQE80L (plasmid) | QIAGEN | | 6x-HIS tag expression vector |
| Recombinant DNA reagent | p601 | Widom lab | | Lowary, P. T. & Widom, J. New DNA sequence rules for high affinity binding to histone octamer and sequence-directed nucleosome positioning. *J. Mol. Biol.* **276**, 19–42 (1998). |
| Sequence-based reagent | 77 w-N | This paper | PCR primers | GTACCCGGGGATCCTCTAGAGTG Ordered from Integrated DNA technologies (IDT) |
| Sequence-based reagent | 0 w-N | This paper | PCR primers | ACAGGATGTATATATCTGACACGTGCC Ordered from Integrated DNA technologies (IDT) |
| Sequence-based reagent | Cy3-0w-N | This paper | PCR primers | Cy3-ACAGGATGTATATATCTGACACGTGCC Ordered from Integrated DNA technologies (IDT) |
| Sequence-based reagent | N-77s | This paper | PCR primers | GATCCTAATGACCAAGGAAAGCATGATTC Ordered from Integrated DNA technologies (IDT) |
| Sequence-based reagent | N-0s | This paper | PCR primers | CTGGAGAATCCCGGTGCCGA Ordered from Integrated DNA technologies (IDT) |
| Sequence-based reagent | N-0s-Cy3 | This paper | PCR primers | Cy3-CTGGAGAATCCCGGTGCCGA Ordered from Integrated DNA technologies (IDT) |
| Peptide, recombinant protein | 3x-FLAG | Sigma-Aldrich | F4799 | |
| Peptide, recombinant protein | Phusion high-fidelity DNA polymerase | New England Biolabs | M0530 | |
| Peptide, recombinant protein | Streptavidin-Eu | Perkin Elmer | 1244-360 | |

*Continued on next page*

*Continued*

| Reagent type (species) or resource | Designation | Source or reference | Identifiers | Additional information |
|---|---|---|---|---|
| Chemical compound | Cy3 maleimide | Lumiprobe | 41080 | |
| Chemical compound | Cy5 maleimide | Lumiprobe | 43080 | |
| Chemical compound | Oregon green 488 maleimide | Thermo Fisher Scientific | O6034 | |
| Commercial assay, kit | Phosphate sensor | Thermo Fisher Scientific | PV4406 | |
| Software | Prism 9 | GraphPad Software, LLC | Version 9.2.0 | |
| Software | cryoSPARC | Structura Biotechnology Inc | Version 4 | Punjani, A., Rubinstein, J. L., Fleet, D. J. & Brubaker, M. A. CryoSPARC: Algorithms for rapid unsupervised cryo-EM structure determination. *Nat. Methods* **14** (2017). |
| Software | ChimeraX | UCSF | | Pettersen, E. F. et al. UCSF ChimeraX: Structure visualization for researchers, educators, and developers. *Protein Sci.* **30** (2021). |
| Other | Planetary ball mill | Retsch | PM-100 | See Method section for SWR1C preparation |
| Other | Mini-prep cell apparatus | Bio-Rad | 1702908 | See Method section for nucleosome reconstitution |

## Yeast strains and culture conditions

The yeast strain *W1588-4C swr1::SWR1– 3xFLAG-P-kanMx-P htz1Δ::natMX* (yEL190) was a gift from Ed Luk (SUNY Stony Brook). Deletion of *SWC5* was constructed by standard gene replacement with a Hygromycin B cassette (pAG32) (*Goldstein and McCusker, 1999*), and used to make *W1588-4C swr1::SWR1– 3xFLAG-P-kanMx-P htz1Δ::natMX swc5Δ::hphMX* (CY2535). The *swc5Δ* strain *MATa his3Δ1 leu2Δ0 met15Δ0 ura3Δ0 YBR231c::kanMX4* (Y03371) used for spot assays was purchased from Euroscarf.

Spot assays were performed with the *swc5Δ* strain (Y03371) transformed with an empty CEN/ARS/URA3 vector pRS416, or ones containing WT and mutant *SWC5*. Cultures were diluted to an $OD_{600}$ of 1 and then serially diluted 1:10. A dilution (7 µl) was spotted onto synthetic complete media lacking uracil with and without 2% formamide. Plates were incubated for 3 days before imaging.

## Plasmid construction

The vector (pQE80L) was used for recombinant Swc5 expression. The *SWC5* gene and *SWC5*[79–303] were inserted in frame with the N-terminal 6x histidine tag through Gibson assembly (*Gibson et al., 2009*) after digesting the vector with BamHI and SphI. Once WT and truncated Swc5 were in frame, the alanine mutants [*swc5* (RRKR-4A) and *swc5* (LDW-3A)] were generated using QuickChange site-directed mutagenesis. The pRS416 *CEN/ARS/URA3* vector was digested with XbaI and SacI to allow for insertion of the *SWC5* gene with 500 bp upstream of the start site and 500 bp downstream of the stop. This region was initially PCR amplified from a W303 strain where the primers had homology to the XbaI and SacI cut sites of the vector. Gibson assembly was used to ligate the vector and insert (CP1566) (*Gibson et al., 2009*). Once the WT *SWC5* gene was inserted, site-directed mutagenesis was used on CP1566 to make *swc5* RRKR-4A (CP1579) and *swc5* LDW-3A (CP1580). *swc5* 79–303 construct was generated using CP1566 to PCR amplify the 500 bp upstream stretch of DNA and start site of *SWC5* and PCR amplifying another DNA fragment starting from Swc5 K79 to 500 bp downstream of the stop. Primers were designed so each PCR fragment had homology to each other and the pRS416 vector at XbaI and SphI so that the two fragments and the pRS416 vector could be ligated together by Gibson assembly. Site-directed mutagenesis was used to generate cysteine mutants used for Oregon Green labeling, as well as for single amino acid substitution used for assembly of acidic patch mutant (AB-apm) dimers – H2A-apm (H2A-E63A, H2A-E65A, H2A-D74A, H2A-D92A, H2A-D93A, H2A-E94A) and H2B-apm (H2B-E109A, H2B-E117A).

## Protein purification for in vitro assays

### Histones

Histones were generated as described previously (*Singh et al., 2019*; *Luger et al., 1999a*). For experiments in this paper, we produced histones H2A, H2B, H2A-K119C, H2A-apm, H2B-apm, H2B-S115C, H2B-apm-S115C, H2A.Z, and H2A.Z-3xFLAG from *S. cerevisiae*, as well as histones H3, H4, and H4-Q27C from *X. laevis*. Fluorescent labels for histones were conjugated on an as needed basis prior to nucleosome or hexasome reconstitution as described previously (*Zhou and Narlikar, 2016*).

### SWR1C

Swr1-3xFlag yeast strains were grown in 6 × 2 l of YEPD, supplemented with adenine at 30°C until reaching an $OD_{600}$ of 3. Cultures were centrifuged and pellets were noodled into liquid nitrogen and stored at −80°C. SWR1C was purified as previously described (*Luk et al., 2010*; *Singh et al., 2019*) with minor modifications. A PM 100 cryomill was used to lyse the harvested yeast noodles with 6 × 1 min cycles at 400 rpm then stored at −80°C. Lysed cell powder (~200 ml) were resuspended in 200 ml 1× HC lysis buffer (25 mM (4-(2-hydroxyethyl)-1-piperazineethanesulfonic acid)HEPES–KOH pH 7.6, 10% glycerol, 1 mM EDTA(Ethylenediaminetetraacetic acid), 300 mM KCl, 0.01% NP40, 10 mM β-glycerophosphate, 1 mM Na-butyrate, 0.5 mM NaF, 1 mM DTT plus 1× protease inhibitors [PI 1 mM PMSF(phenylmethylsulfonyl fluoride), 1 mM benzamidine, 0.1 mg/ml pepstatin A, 0.1 mg/ml leupeptin, 0.1 mg/ml chymostatin]) in a 1 l beaker with a stir bar at 4°C. Thawed cells were centrifuged for 2 hr at 35,000 rpm in a Ti45 rotor at 4°C then whole-cell extract was transferred to a 250-ml falcon tube along with anti-Flag M2 agarose resin (1 ml bed volume) and nutated at 4°C for 4 hr. Resin and extract mixture was transferred in 25 ml increments to a 25-ml gravity column. After flow through of extract, the resin was washed with 4 × 25 ml B-0.5 buffer (25 mM HEPES, pH 7.6, 1 mM EDTA, 2 mM MgCl₂, 10 mM β-glycerophosphate, 1 mM Na-butyrate, 0.5 mM NaF, 500 mM KCl, 10% glycerol, 0.01% NP40, and 1× PI) followed by 3 × 10 ml B-0.1 buffer (25 mM HEPES, pH 7.6, 1 mM EDTA, 2 mM MgCl₂, 10 mM β-glycerophosphate, 1 mM Na-butyrate, 0.5 mM NaF, 100 mM KCl, 10% glycerol, 0.01% NP40, and 1× PI). For purification of SWR1C^swc5Δ where recombinant Swc5 is added back, rSwc5 would be diluted to 5 μM in 2 ml of B-0.1 and added to resin for 15 min at 4°C followed by 2 × 5 ml B-0.5 washes then 2 × 5 ml B-0.1 washes. SWR1C was eluted by nutating resin in 1 ml B-0.1 with 0.5 mg/ml recombinant 3xFlag peptide (Genscript) for 1 hr twice in series. Collected eluant was combined and concentrated using a 100-kDa cutoff Amicon Ultra-0.5 ml centrifugal filter (Millipore). After concentrating down to ~150 μl, 3xFlag peptide is removed by serial dilution with fresh B-0.1 four more times in the concentrator. The subsequently concentrated SWR1C was aliquoted, flash frozen, and stored at −80°C. SWR1C concentration was determined by sodium dodecyl sulfate–polyacrylamide gel electrophoresis (SDS–PAGE) using a BSA(bovine serum albumin) (NEB) standard titration followed by SYPRO Ruby (Thermo Fisher Scientific) staining and quantification using ImageQuant 1D gel analysis.

### Recombinant Swc5

Swc5 and variants were first cloned into a pQE80L expression vector in frame with a six histidine N-Terminal tag through Gibson assembly. Plasmids were transformed into Rosetta 2 DE3 competent cells (Novagen). 1 l cultures of 2× YT (Yeast extract, tryptone) were grown at 37°C to $OD_{600}$ of 0.5–0.7 before adding 0.4 mM IPTG(Isopropyl β-d-1-thiogalactopyranoside). After adding IPTG, cells continued to shake at room temperature for 3 hr. Cells were harvested by centrifugation at 4°C and 1 l of Swc5 bacterial pellet was resuspended in 20 ml lysis buffer (10 mM imidazole, 50 mM Tris pH 8.0, 300 mM NaCl, 1 mM PMSF, 1 mM DTT) before being flash frozen in liquid nitrogen and stored at −80°C. Pellets were thawed in a 37°C water bath then fresh 0.5 mM PMSF and 1 mM benzamidine were added. Cells were sonicated 5 × 30 s at 15% power, incubating on ice for 1 min between sonication intervals. Lysate was then centrifuged at 14,000 rpm in a JA-17 rotor at 4°C for 25 min. Whole-cell extract was incubated with 800 μl of HisPur Ni-NTA resin slurry, that had been prewashed with lysis buffer, for 2 hr at 4°C on nutator. The resin and whole-cell extract mixture was centrifuged at 750 rcf for 5 min. The supernatant was aspirated, and the resin was placed in a 25-ml gravity column and washed with 5 × 10 ml wash buffer (500 mM NaCl, 10 mM imidazole, 50 mM Tris pH 8.0). Swc5 was eluted by capping the gravity column and adding 400 μl elution buffer (500 mM NaCl, 300 mM imidazole, 50 mM Tris pH 8.0) to resin and incubating at 4°C for 10 min then collecting the flow through.

The Swc5 eluate was dialyzed against 2 × 250 ml of storage buffer (50 mM NaCl, 50 mM Tris pH 7.4, 10% glycerol) then flash frozen and stored at −80°C. Concentration was determined by BSA standard on an SDS–PAGE gel.

## Nucleosome reconstitution

These experiments utilized a variety of nucleosomes, both in terms of histone variants, labels, and mutations. Broadly, all nucleosomes contained a tetramer of *X. laevis* H3 and H4, along with *S. cerevisiae* H2A/H2B heterodimers, with mutations or variant substitutions as appropriate, reconstituted on DNA containing the Widom 601 positioning sequence with or without linkers and/or fluorescent labels. We refer to the Widom 601 sequence as a 147-bp element, though structural studies have indicated that this sequence assembles a 145-bp nucleosome core particle (*Vasudevan et al., 2010*). We decided to use *X. laevis* H3/H4 tetramers due to slightly increased stability of nucleosomal product in solution. These histones show high sequence conservation with *S. cerevisiae* H3 and H4 and we have previously demonstrated there is no difference in SWR1C eviction activity on full yeast or hybrid yeast-xenopus nucleosomes (*Singh et al., 2019*). Furthermore, the H4 tail region that interacts with Swc5 in our cryo-EM models is virtually identical in the orthologs.

For experiments using nucleosomes with symmetrical dimers, hybrid octamers were generated as previously described and nucleosomes were reconstituted via salt dialysis deposition on a Widom 601 positioning sequence containing DNA template (*Lowary and Widom, 1998*; *Luger et al., 1999b*).

Symmetrical nucleosomes produced include AB/AB and AB-apm/AB-apm mutant nucleosomes used for Swc5 quenching and gel shift assays contained yH2B-S115C or xH4-Q27C labeled with Oregon Green 488 Malemide (Thermo Fisher) on a 0N0 DNA template (*Lowary and Widom, 1998*). Additionally, AB/AB and AB-apm/AB-apm containing nucleosomes in which all H2A dimers contain the label Cy5 were assembled on 77N0-Cy3 and 0N0-Cy3 templates.

Asymmetrical nucleosomes were produced by first reconstituting *X. laevis* H3/H4 tetramers as well as *S. cerevisiae* dimers of interest separately instead of generating octamers. Tetramers, a single dimer type, and template DNA were mixed in a 1.4:1.8:1.0 ratio followed by reconstitution via salt dialysis identically to regular nucleosome reconstitution. This ratio was found to reliably produce a mixture of hexasomes and nucleosomes, and in the case of hexasomes the dimer is known to be deposited on the 'strong' side of the Widom position sequence (*Levendosky et al., 2016*). For asymmetric nucleosomes containing a single dimer labeled with Cy5, the template DNA was varied such that the labeled dimer is deposited on the strong side of the 601 positioning sequence. Note that our previous study demonstrated that the rates of H2A.Z deposition by SWR1C are nearly identical for either the weak or the strong side of a 601 mononucleosome (*Lowary and Widom, 1998*). Hexasomes are then purified using a Bio-Rad MiniPrep Cell apparatus, a method by which proteins are collected via fraction collector after passage through a native-PAGE tube gel (60:1 Acrylamide/Bis), which is sufficient to separate hexasomes from nucleosomes (*Levendosky et al., 2016*; *Levendosky and Bowman, 2019*; *Nodelman et al., 2020*). After checking fractions on a native-PAGE gel (29:1 Acrylamide/Bis) stained with SYBR Gold, the hexasome containing fractions were pooled and concentrated, followed by buffer exchange into remodeling buffer (25 mM HEPES, pH 7.6, 0.2 mM EDTA, 5 mM MgCl$_2$, 70 mM KCl, 1 mM DTT) with 20% glycerol, aliquoting, and flash-freezing. The introduction of a *S. cerevisiae* heterodimer to hexasomes in solution leads to spontaneous incorporation and formation of a nucleosome, allowing for production of asymmetric nucleosomes with known orientation.

Six types of asymmetric nucleosomes for dimer exchange assays were produced in this manner containing linker-proximal or distal H2A-Cy5/H2B (AB-Cy5) heterodimers with placement of unlabeled AB, ZB, or AB-apm dimers on the contralateral side on a 77N0-Cy3 template DNA. Additionally, a linker distal AB-apm-Cy5 and linker proximal AB nucleosome, as well as AB/AB-Cy5 and ZB/AB-Cy5 core particles (0N0 template DNA) were produced.

For FP and ATPase studies, unlabeled dimers were used to produce hexasomes containing linker distal AB or AB-apm dimers on 77N0-Cy3. These hexasomes were then used to generate nucleosomes with dimers (listed here and throughout the manuscript in linker proximal/linker distal order) AB/AB, AB/AB-apm, AB-apm/AB, or AB-apm/AB-apm, all on the 77N0-Cy3 template.

## Nucleosome dimer exchange assays

FRET-based dimer exchanges assays were performed as previously described (*Singh et al., 2019*) using an ISS PC1 spectrofluorometer, a Tecan Infinite M1000 PRO microplate reader, or a Tecan Spark microplate reader. The nucleosome remodeling reactions (50–100 µl) were performed in remodeling buffer at room temperature with 30–60 nM SWR1C, 10 nM 77N0-Cy3 and at least one dimer-Cy5 containing nucleosome, 60–70 nM ZB dimers, and 1 mM ATP (to start reaction). Dimer exchange was observed by exciting the Cy3 fluorophore at 530 nm and monitoring the Cy5 emission 670. Data for nucleosomes with both dimers labeled were fit to the two-phase decay equation (*Equation 1*; where $Y$ denotes RFU, $Y_i$ the initial RFU, $Y_p$ the final RFU, $P_f$ the fast phase expressed as a fraction, $P_s$ the slow phase expressed as a fraction, $k_f$ the rate constant of the fast phase, $k_s$ the rate constant of the slow phase, and $t$ is time) while data for reactions with only one dimer labeled were fit to the one-phase decay equation (*Equation 2*; where $Y$ denotes RFU, $Y_i$ the intial RFU, $Y_p$ the final RFU, $k$ is the rate constant, and $t$ is time).

$$Y = \left(P_f\left(Y_i - Y_p\right)\right)^{-k_f t} + \left(P_s\left(Y_i - Y_p\right)\right)^{-k_s t} + Y_p \tag{1}$$

$$Y = \left(Y_i - Y_p\right)^{-kt} + Y_p \tag{2}$$

Gel-based dimer exchange assays used a modified protocol from a previous study (*Mizuguchi et al., 2004*; *Ranjan et al., 2013*). A 150-µl dimer exchange reaction contained 30 nM SWR1C, 10 nM 77N0 nucleosomes (WT or APM), and 70 nM H2A.Z$^{flag}$/H2B in remodeling buffer with 0.1 mg/ml BSA. The reaction was performed at room temperature and 20 µl of the reaction was taken at each time point (time 0 is before 1 mM ATP is added) and quenched with 1 µg of plasmid DNA to separate SWR1C and dimers from the nucleosome. Each time point was stored on ice until last sample was quenched. Each time point was loaded onto a 6% 0.5× TBE Native PAGE gel (29:1 Acrylamide/Bis) and electrophoresed for 90 min at 120 V. Gels were stained with SYBR gold and imaged on a GE Typhoon.

## FP assay

FP assays were performed as previously described (*Baier and Peterson, 2022*) using a CLARIOstar microplate reader. The affinity of SWR1C for various nucleosomal substrates was tested by mixing 10 nM 77N0-Cy3 nucleosomes with SWR1C serially diluted in concentration from approximately 1 µM to 1 nM in remodeling buffer. The dimensionless value of polarization was plotted for each concentration of SWR1C and the Morrison equation (*Equation 3*; where *Bmax* is maximum observed binding, $[F]$ is the concentration of the labeled nucleosomes, $[E]$ is the concentration of SWR1C, and $K_{d,app}$ is the apparent dissociation constant) was used determine the binding affinity for substrates where binding was observed.

$$Y = Bmax\left(\frac{\left([F] + [E] + K_{d,app}\right) - \sqrt{\left([F] + [E] + K_{d,app}\right)^2 - 4[F][E]}}{2[F]}\right) \tag{3}$$

## ATPase assay

ATPase activity was measured using a phosphate-binding protein reporter assay. For the phosphate sensor assay (*Luk et al., 2010*; *Brune et al., 1994*), 50 µl reactions were prepared with 8 nM SWR1C mixed with 0.5 µM phosphate sensor (fluorophore-conjugated phosphate-binding protein, Thermo Fisher) in remodeling buffer. Reactions containing nucleosome and ZB dimers had concentrations of 20 and 30 nM, respectively. Reactions were initiated by adding 100 µM ATP and monitored by the measuring the fluorescence of the MDCC fluorophore on a Tecan Spark by exciting at 430 nm and setting the emission filter to 450 nm. All reagents and the microplate were pre-treated with a 'Pi Mop' to remove free phosphate as previously described (*Luk et al., 2010*; *Brune et al., 1994*). FI was monitored over time and the linear range of intensity change was plotted to calculate the rate in reciprocal seconds.

## Swc5-binding assays

Gel shifts with recombinant WT or Swc5 derivatives were performed in 15 µl reactions. Reactions contained varying amounts of Swc5 in a total volume of 5 µl of storage buffer (10% glycerol, 50 mM Tris pH 7.4, 50 mM NaCl), 0.25 µl 300 nM 0N0 nucleosomes, 6.15 µl water, 0.6 µl 50% glycerol, and 3 µl 5× binding buffer (125 mM HEPES pH 7.3, 250 mM NaCl, 25 mM $MgCl_2$, 0.05% Tween 20, 0.5 mg/ml BSA, 5 mM DTT). Reaction incubated in the dark at room temp for 30 min. Samples were loaded onto 6% native-PAGE gels (29:1 Acrylamide/Bis) and electrophoresed at 230 V for 1 hr. Gels were imaged on a GE typhoon imager and binding was calculated by measuring the disappearance of the free nucleosome band. GraphPad Prism 8 was used to generate binding curves.

Swc5 fluorescent quenching assays were assembled similar to the gel shift assays but in 40 µl reactions with 10 nM 0N0 Oregon Green nucleosomes (8 µl 5× binding buffer, 1.3 µl 300 nM nucleosomes, 11.7 µl water, 19 µl Swc5/storage buffer). Master mixes of each reaction incubated for 30 min at room temp in the dark before being loaded into 384-well flat black plate (PerkinElmer) and scanning each well 20 times on a Tecan Spark plate reader at 488 nm excitation and 530 nm emission. Quenching was calculated by taking the average relative fluorescence units (RFUs) of each Swc5 containing well and dividing RFU average of the nucleosome only wells. The normalized fluorescence was plotted using Prism 8.

Time-resolved FRET nucleosome-binding assays were performed as described by *Wesley et al., 2022*. Acceptor mixtures were prepared by mixing ULight alpha-6xHIS acceptor antibody (PerkinElmer) with 6xHIS-tagged Swc5 variants at a ratio of 1:20 and serially diluting across 13 concentrations in H66 buffer (20 mM HEPES pH 7.5, 66 mM NaCl, 5 mM MgCl2, 5 mM DTT, 5% glycerol, 0.01% NP-40, 0.01% CHAPS, and 100 µg/ml BSA). 2× donor mixtures were prepared by mixing 4 nM streptavidin-Eu (PerkinElmer) with or without 2 nM yeast nucleosomes containing 177 bp of Widom 601 DNA (31 + 145 + 1) with a 5′ biotin group on the 31 bp extension and incubating for 30 min at room temperature. Samples were prepared in 384-well plates by mixing 5 µl of 2× donor mixtures with 5 µL of acceptor mixtures at each dilution. Fluorescence signals were acquired at room temperature in a Victor Nivo multimode fluorescent plate reader (PerkinElmer) using an excitation filter at 320 nm and emission filters at 615 and 665 nm. Emission signals at 615 and 665 nm were measured simultaneously following a 100 µs delay. Kd values were determined from triplicate titrations of each Swc5 variant and are reported as means ± standard error of the mean.

## Cryo-EM sample preparation

Swc5[79–303] was mixed with 147 bp Widom 601 sequence-containing nucleosomes at a 2.5:1 molar ratio in H50 buffer (10 mM HEPES pH 7.5, 50 mM NaCl, 1 mM DTT(Dithiothreitol)). Precipitated material was removed by centrifugation at 16,000 × *g* for 10 min. The sample was chemically cross-linked using the GraFix method in a 10–30% glycerol gradient (*Stark, 2010*), prepared with a Gradient Master instrument (Biocomp) from top (H50 + 10% vol/vol glycerol) and bottom (H50 + 30% vol/vol glycerol, 0.15% vol/vol glutaraldehyde) solutions. The sample was applied on top of the gradient and centrifuged in a SW40Ti rotor (Beckman-Coulter) at 35,000 rpm, 4°C for 16 hr. Fractions were collected manually by pipetting from the top, and the reaction was quenched by adding Tris pH 7.5 to each fraction to 50 mM. Fractions were analyzed using a 6% native gel (29:1 Acrylamide/Bis) stained with ethidium bromide and negative-stain EM with uranyl formate to select for monodisperse particles of the complex. Selected fractions were dialyzed into H50 buffer supplemented with 0.1 mM PMSF and concentrated to ~1 mg/ml.

## Cryo-EM data collection and analysis

Swc5/nucleosome complex was prepared at 0.9 mg/ml. Quantifoil R1.2/1.3 holey carbon, 300 mesh, copper grids (Electron Microscopy Sciences) were glow discharged for 45 s at 15 mA with an easiGlow device (PELCO). After glow discharge, 3 µl of sample was pipetted onto the grid and Grade 595 filter paper (Ted Pella) was used to blot the grid for 3 s, at 4°C and 100% humidity. The grid was plunge-frozen in liquid ethane cooled to the temperature of liquid nitrogen with a Vitrobot Mark IV instrument (Thermo Scientific). A Krios G3i microscope (Thermo Scientific) operating at 300 kV was used in conjunction with a K3 direct electron detector camera (Gatan) to collect data in super-resolution mode at the Pacific Northwest Cryo-EM Center. 13,855 movies were collected at a nominal magnification of 29,000× with a physical pixel size of 0.788 Å (super-resolution, 0.394 Å). The total dose per exposure

was 50 e⁻/Å² at a dose rate of ~18.5 e⁻/pix/s and each exposure was fractioned into 50 subframes. Data were recorded in the defocus range of −0.8 to −2.8 μm.

Data were processed with the cryoSPARC v4 suite (*Figure 7—figure supplement 1*; *Punjani et al., 2017*). Super-resolution movies were motion-corrected with 'Patch Motion Correction' and binned using 2× Fourier cropping. CTF(contrast transfer function) values were estimated with 'Patch CTF Estimation'. Initial particles were picked with 'Blob Picker' from a subset of micrographs, then used for ab initio 3D reconstruction, and the best obtained model was used to generate templates for particle picking with 'Template Picker'. Template-picked particles were refined into three classes using 'Heterogenous Refinement' with the previously generated ab initio reconstructions as initial volumes to remove junk particles. 1,846,495 nucleosomal particles were further filtered to remove low-quality particles using the 'random-phase 3D classification' strategy (*Gong et al., 2016*). Briefly, particles were sequentially subjected to multiple rounds of Heterogeneous Refinement against a low-pass filtered reconstruction (40, 30, 25, 20, 15, 10, and 8 Å low-pass) of the output volume from the previous round. This strategy yielded two classes of high-quality particles with the nucleosomal DNA either (1) completely wrapped around the core (585,573 particles, Non-uniform Refinement to 3.0 Å resolution), or (2) partially peeling off (392,348 particles, 3.1 Å). The Swc5 moiety is extremely noisy in these reconstructions due to apparent structural heterogeneity. We tried employing various methods including '3D Variability Analysis' (*Punjani and Fleet, 2021*) to either further classify the particles into more homogenous subsets or to describe the intrinsic conformational dynamics, but with limited success (data not shown). We obtained the most informative results with cryoSPARC's algorithm of '3D Classification' without alignment. The 585 k particles of fully wrapped nucleosome were classified into 10 classes using a mask focused on the sides of the histone core in the regions with Swc5 moiety apparent. The obtained particle classes were refined to ~3.1 Å using 'Non-uniform Refinement' (*Punjani et al., 2020*). Modeling was performed with UCSF ChimeraX (*Pettersen et al., 2021*).

## Acknowledgements

We thank Ed Luk (SUNY Stony Brook) for the kind gift of the Swr1-3xFLAG strain; Jessica Feldman for vector pQE80L, and members of the Peterson lab for helpful discussions. We thank Joseph Cho for preparing cryo-EM grids and collecting preliminary data at the Penn State cryo-EM facility, and we thank Jose M Espinola-Lopez for assistance with the TR-FRET assays. This work was supported by the National Institutes of Health [R35-GM122519 to CLP] and [R35GM127034 to ST], and by the Estonian Research Council [PUTJD906 to PE]. A portion of this research was supported by NIH grant U24GM129547 and performed at the PNCC at OHSU and accessed through EMSL (grid.436923.9), a DOE Office of Science User Facility sponsored by the Office of Biological and Environmental Research. This project is funded, in part, under a grant from the Pennsylvania Department of Health using Tobacco CURE Funds. The Department specifically disclaims responsibility for any analyses, interpretations, or conclusions. Molecular graphics figures were generated using UCSF ChimeraX. UCSF ChimeraX is developed by the Resource for Biocomputing, Visualization, and Informatics at the University of California, San Francisco, with support from NIH R01-GM129325 and the Office of Cyber Infrastructure and Computational Biology, National Institute of Allergy and Infectious Diseases.

## Additional information

### Funding

| Funder | Grant reference number | Author |
| --- | --- | --- |
| National Institutes of Health | R35-GM122519 | Craig L Peterson |
| National Institutes of Health | R35GM127034 | Song Tan |
| Estonian Research Council | PUTJD906 | Priit Eek |

The funders had no role in study design, data collection, and interpretation, or the decision to submit the work for publication.

## Author contributions
Alexander S Baier, Conceptualization, Data curation, Formal analysis, Investigation, Methodology, Writing – original draft, Writing – review and editing; Nathan Gioacchini, Data curation, Formal analysis, Investigation, Methodology, Writing – original draft; Priit Eek, Conceptualization, Data curation, Formal analysis, Investigation, Methodology; Erik M Leith, Data curation, Investigation, Methodology; Song Tan, Conceptualization, Data curation, Supervision, Funding acquisition, Writing – original draft, Project administration, Writing – review and editing; Craig L Peterson, Conceptualization, Formal analysis, Supervision, Funding acquisition, Investigation, Writing – original draft, Project administration, Writing – review and editing

## Author ORCIDs
Alexander S Baier ⓘ https://orcid.org/0000-0003-1647-9477
Craig L Peterson ⓘ http://orcid.org/0000-0002-9448-555X

Reviewer #1 (Public Review): https://doi.org/10.7554/eLife.94869.3.sa1
Author response https://doi.org/10.7554/eLife.94869.3.sa2

# Additional files

## Supplementary files
• MDAR checklist

## Data availability
Source data are provided with this paper. The cryo-EM maps depicted in Figure 6A–D have been deposited to the Electron Microscopy Data Bank under accession codes EMD-41852, EMD-41839, EMD-41853, and EMD-41851, respectively. The raw micrographs, extracted particles, and final 3D classification subsets are available from the Electron Microscopy Public Image Archive with the accession code EMPIAR-11681.

The following datasets were generated:

| Author(s) | Year | Dataset title | Dataset URL | Database and Identifier |
|---|---|---|---|---|
| Eek P, Tan S | 2024 | Cryo-EM structure of yeast SWR1C subunit Swc5 bound to the nucleosome, 3D class 0 | https://www.ebi.ac.uk/emdb/EMD-41839 | Electron Microscopy Data Bank, EMD-41839 |
| Eek P, Tan S | 2024 | Cryo-EM structure of yeast SWR1C subunit Swc5 bound to the nucleosome, 3D class 1 | https://www.ebi.ac.uk/emdb/EMD-41851 | Electron Microscopy Data Bank, EMD-41851 |
| Eek P, Tan S | 2024 | Cryo-EM structure of yeast SWR1C subunit Swc5 bound to the nucleosome, 3D class 2 | https://www.ebi.ac.uk/emdb/EMD-41852 | Electron Microscopy Data Bank, EMD-41852 |
| Eek P, Tan S | 2024 | Cryo-EM structure of yeast SWR1C subunit Swc5 bound to the nucleosome, 3D class 4 | https://www.ebi.ac.uk/emdb/EMD-41853 | Electron Microscopy Data Bank, EMD-41853 |
| Eek P, Tan S | 2024 | Cryo-EM structure of yeast SWR1C subunit Swc5 bound to the nucleosome | https://www.ebi.ac.uk/empiar/EMPIAR-11681 | Electron Microscopy Public Image Archive, EMPIAR-11681 |

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
