## [Editor Report · eLife assessment]

This manuscript presents an **important** analysis of the role that the nucleosome acidic patch plays in SWR1-catalyzed histone exchange. This manuscript contains **convincing** data which significantly expands our understanding of the complex process of H2A.Z deposition by SWR1 and therefore would be of interest to a broad readership.

---

## [Referee Report · Reviewer #1 (Public Review)]

This manuscript presents an extremely exciting and very timely analysis of the role that the nucleosome acidic patch plays in SWR1-catalyzed histone exchange. Intriguingly, SWR1 loses activity almost completely if any of the acidic patches are absent. To my knowledge, this makes SWR1 the first remodeler with such a unique and pronounced requirement for the acidic patch. The authors demonstrate that SWR1 affinity is dramatically reduced if at least one of the acidic patches is absent, pointing to a key role of the acidic patch in SWR1 binding to the nucleosome. The authors also pinpoint a specific subunit - Swc5 - that can bind nucleosomes and engage the acidic patch and obtain a cryo-EM structure of Swc5 bound to a nucleosome. They also identify a conserved arginine-rich motif in this subunit that is critical for nucleosome binding and histone exchange in vitro and for SWR1 function in vivo. The authors provide evidence that suggests a direct interaction between this motif and the acidic patch.

Strengths:

The manuscript is well-written and the experimental data are of outstanding quality and importance for the field. This manuscript significantly expands our understanding of the fundamentally important and complex process of H2A.Z deposition by SWR1 and would be of great interest for a broad readership.

---

## [Author Response]

The following is the authors’ response to the original reviews.

The reviewer comments have been helpful, and we have revised the manuscript to address the concerns of reviewer 2. In addition to text changes, we also added a negative control to Figure 1 to address concerns about photobleaching or DNA unwrapping.

**Reviewer #1:**
This manuscript presents an extremely exciting and very timely analysis of the role that the nucleosome acidic patch plays in SWR1-catalyzed histone exchange. Intriguingly, SWR1 loses activity almost completely if any of the acidic patches are absent. To my knowledge, this makes SWR1 the first remodeler with such a unique and pronounced requirement for the acidic patch. The authors demonstrate that SWR1 affinity is dramatically reduced if at least one of the acidic patches is absent, pointing to a key role of the acidic patch in SWR1 binding to the nucleosome. The authors also pinpoint a specific subunit - Swc5 - that can bind nucleosomes, engage the acidic patch, and obtain a cryo-EM structure of Swc5 bound to a nucleosome. They also identify a conserved arginine-rich motif in this subunit that is critical for nucleosome binding and histone exchange in vitro and for SWR1 function in vivo. The authors provide evidence that suggests a direct interaction between this motif and the acidic patch.Strengths:The manuscript is well-written and the experimental data are of outstanding quality and importance for the field. This manuscript significantly expands our understanding of the fundamentally important and complex process of H2A.Z deposition by SWR1 and would be of great interest to a broad readership.

We thank the reviewer for their enthusiastic and positive comments on our work.

**Reviewer #2:**
Summary:In this study, Baier et al. investigated the mechanism by which SWR1C recognizes nucleosomal substrates for the deposition of H2A.Z. Their data convincingly demonstrate that the nucleosome's acidic patch plays a crucial role in the substrate recognition by SWR1C. The authors presented clear evidence showing that Swc5 is a pivotal subunit involved in the interaction between SWR1C and the acidic patch. They pared down the specific region within Swc5 responsible for this interaction. However, two central assertions of the paper are less convincing. First, the data supporting the claim that the insertion of one Z-B dimer into the canonical nucleosome can stimulate SWR1C to insert the second Z-B dimer is somewhat questionable (see below). Given that this claim contradicts previous observations made by other groups, this hypothesis needs further testing to eliminate potential artifacts. Secondly, the claim that SWR1C simultaneously recognizes the acidic patch on both sides of the nucleosome also needs further investigation, as the assay used to establish this claim lacks the sensitivity necessary to distinguish any difference between nucleosomal substrates containing one or two intact acidic patches.Strengths:As mentioned in the summary, the authors presented clear evidence demonstrating the role of Swc5 in recognition of the nucleosome acidic patch. The identification of the specific region in Swc5 responsible for this interaction is important.

We thank the reviewer for their careful critique of our work. Below we address each major concern.

Major comments:(1) Figure 1B: It is unclear how much of the decrease in FRET is caused by the bleaching of fluorophores. The authors should include a negative control in which Z-B dimers are omitted from the reaction. In the absence of ZB dimers, SWR1C will not exchange histones. Therefore, any decrease in FRET should represent the bleaching of fluorophores on the nucleosomal substrate, allowing normalization of the FRET signal related to A-B eviction.

In this manuscript, as well as in our two previous publications (Singh et al., 2019; Fan et al.,2022), we have presented the results of no enzyme controls, +/- ZB dimers, no ATP controls, or AMP-PNP controls for our FRET-based, H2A.Z deposition assay (see also Figure S3). We do not observe significant levels of photobleaching in this assay, either during ensemble measurements or in an smFRET experiment. To aid the reader, we have added the AMP-PNP data for the experiment shown in Figure 1B. The results show there is less than a 10% decrease in FRET over 30’, and the signal from the double acidic patch disrupted nucleosome is identical to this negative control.

(2) Figure S3: The authors use the decrease in FRET signal as a metric of histone eviction. However, Figure S3 suggests that the FRET signal decrease could be due to DNA unwrapping. Histone exchange should not occur when SWR1C is incubated with AMP-PNP, as histone exchange requires ATP hydrolysis (10.7554/eLife.77352). And since the insertion of Z-B dimer and the eviction of A-B dimer are coupled, the decrease of FRET in the presence of AMP-PNP is unlikely due to histone eviction or exchange. Instead, the FRET decrease is likely due to DNA unwrapping (10.7554/eLife.77352). The authors should explicitly state what the loss of FRET means.

We agree with the reviewer, that loss of FRET can be due to DNA unwrapping from the nucleosome. We have previously demonstrated this activity by SWR1C in our smFRET study (Fan et al., 2022). However, DNA unwrapping is highly reversible and has a time duration of only 1-3 seconds. We and others have not observed stable unwrapping of nucleosomes by SWR1C, but rather the stable loss of FRET reports on dimer eviction. We assume the reviewer is concerned about the rather large decrease in FRET signal shown in the AMP-PNP controls for Figure S3, panels A and D. For the other 7 panels, the decrease in FRET with AMP-PNP are minimal. In fact, if we average all of the AMP-PNP data points, the rate of FRET loss is not statistically different from no enzyme control reactions (nucleosome plus ZB dimers).

Data for panels A and D used a 77NO nucleosomal substrate, with Cy3 labeling the linker distal dimer. This is our standard DNA fragment, and it was used in Figure 1B. The only difference between data sets is that the data shown in Fig 1B used nucleosome reconstituted with a Cy5-labelled histone octamer, rather than the hexasome assembly method used for Fig S3. Three points are important. First, for all of these substrates, we assembled 3 independent nucleosomes, and the results are highly reproducible. Two, we performed a total of 6 experiments for the 77NO-Cy5 substrates to ensure that the rates were accurate (+/-ATP). Third, and most important, we do not see this decrease in FRET signal in the absence of SWR1C (no enzyme control). This data was included in the data source file. Thus, it appears that there is significant SWR1C-induced nucleosome instability for these two hexasome-assembled substrates. We now note this in the legend to Figure S3. Key for this work, however, is that there is a large increase in the rate of FRET loss in the presence of ATP, and this rate is faster when a ZB dimer was present at the linker proximal location. In response to the last point, we state in the first paragraph of the results: “The dimer exchange activity of SWR1C is monitored by following the decrease in the 670 nm FRET signal due to eviction of the Cy5-labeled AB-Cy5 dimer (Figure 1A).”

(3) Related to point 2. One way to distinguish nucleosomal DNA unwrapping from histone dimer eviction is that unwrapping is reversible, whereas A-B eviction is not. Therefore, if the authors remove AMP-PNP from the reaction chamber and a FRET signal reappears, then the initial loss of FRET was due to reversible DNA unwrapping. However, if the removal of AMP-PNP did not regain FRET, it means that the loss of FRET was likely due to A-B eviction. The authors should perform an AMP-PNP and/or ATP removal experiment to make sure the interpretation of the data is correct.

See response to item 2 above

(4) The nature of the error bars in Figure 1C is undefined; therefore, the statistical significance of the data is not interpretable.

We apologize for not making this more explicit for each figure. The error bars report on 95% confidence intervals from at least 3 sets of experiments. This statement has been added to the legend.

(5) The authors claim that the SWR1C requires intact acidic patches on both sides of the nucleosomes to exchange histone. This claim was based on the experiment in Figure 1C where they showed mutation of one of two acidic patches in the nucleosomal substrate is sufficient to inhibit SWR1C-mediated histone exchange activity. However, one could argue that the sensitivity of this assay is too low to distinguish any difference between nucleosomes with one (i.e., AB/AB-apm) versus two mutated acidic patches (i.e., AB-apm/AB-apm). The lack of sensitivity of the eviction assay can be seen when Figure 1B is taken into consideration. In the gel-shift assay, the AB-apm/AB-apm nucleosome exhibited a 10% SWR1C-mediated histone exchange activity compared to WT. However, in the eviction assay, the single AB/AB-apm mutant has no detectable activity. Therefore, to test their hypothesis, the authors should use the more sensitive in-gel histone exchange assay to see if the single AB/AB-apm mutant is more or equally active compared to the double AB-apm/AB-apm mutant.

Our pincher model is based on three, independent sets of data, not just Figure 1C. First, as noted by the reviewer, we find that disruption of either acidic patch cripples the dimer exchange activity of SWR1C in the FRET-based assay. Whether the defect is identical to that of the double APM mutant nucleosome does not seem pertinent to the model. In a second set of assays, we used fluorescence polarization to quantify the binding affinity of SWR1C for wildtype nucleosomes, a double APM nucleosome, or each single APM nucleosome. Consistent with the pincher model, each single APM disruption decreases binding affinity at least 10-fold (below the sensitivity of the assay). Finally, we monitored the ability of different nucleosomes to stimulate the ATPase activity of SWR1C. Consistent with the pincher model, a single APM disruption was sufficient to eliminate nucleosome stimulation.

(6) The authors claim that the AZ nucleosome is a better substrate than the AA nucleosome. This is a surprising result as previous studies showed that the two insertion steps of the two Z-B dimers are not cooperative (10.7554/eLife.77352 and 10.1016/J.CELREP.2019.12.006). The authors' claim was based on the eviction assay shown in Fig 1C. However, I am not sure how much variation in the eviction assay is contributed by different preparations of nucleosomes. The authors should use the in-gel assay to independently test this hypothesis.

For all data shown in our manuscript, at least three different nucleosome preparations were used. The impact of a ZB dimer on the rates of dimer exchange was highly reproducible among different nucleosome preparations and experiments. We also see reproducible ZB stimulation for three different substrates – with ZB on the linker proximal side, the linker distal side, and on one side of a core particle. We do not believe that our data are inconsistent with previous studies. First, the previous work referenced by the reviewer performed dimer exchange reactions with a large excess of nucleosomes to SWR1C (catalytic conditions), whereas we used single turnover reactions. Secondly, our study is the first to use a homogenous, ZA heterotypic nucleosome as a substrate for SWR1C. All previous studies used a standard AA nucleosome, following the first and second rounds of dimer exchange that occur sequentially. And finally, we observe only a 20-30% increase in rate by a ZB dimer (e.g. 77N0 substrates), and such an increase was unlikely to have been detected by previous gel-based assays.

Minor comments:(1) Abstract line 4: To say 'Numerous' studies have shown acidic patch impact chromatin remodeling enzymes activity may be too strong.

Removed

(2) Page 15, line 15: The authors claim that swc5∆ was inviable on formamide media. However, the data in Figure 8 shows cell growth in column 1 of swc5∆.

The term ‘inviable’ has been replaced with ‘poor’ or ‘slow growth’

(3) The authors should use standard yeast nomenclature when describing yeast genes and proteins. For example, for Figure 8 and legend, Swc5∆ was used to describe the yeast strain BY4741; MATa; his3Δ1; leu2Δ0; met15Δ0; ura3Δ0; YBR231c::kanMX4. Instead, the authors should describe the swc5∆ mutant strain as BY4741 MAT a his3∆1 leu2∆0 met15∆0 ura3∆0 swc5∆::kanMX4. Exogenous plasmid should also be indicated in italics and inside brackets, such as [SWC5-URA3] or [swc5(R219A)-URA3].

We apologize for missing this mistake in the Figure 8 legend. We had inadvertently copied this from the euroscarf entry and forgot to edit the entry. We decided not to add all the plasmid names to the figure, as it was too cluttered. We state in the figure legend that the panels show growth of swc5 deletion strains harboring the indicated swc5 alleles on CEN/ARS plasmids.

(4) According to Lin et al. 2017 NAR (doi: 10.1093/nar/gkx414), there is only one Swc5 subunit per SWR1C. Therefore, the pincher model proposed by the authors would suggest that there is a missing subunit that recognizes the second acidic patch. The authors should point out this fact in the discussion. However, as mentioned in Major comment 6, I am not sure if the pincer model is substantiated.

In our discussion, we had noted that the published cryoEM structure had suggested that the Swc2 subunit likely interacts with the acidic patch on the dimer that is not targeted for replacement, and we proposed that Swc5 interacts with the acidic patch on the exchanging H2A/H2B dimer. We have now made this more clear in the text.